

# Projecting management-relevant change of undeveloped coastal barriers with the Mesoscale Explicit Ecogeomorphic Barrier model (MEEB) v1.0

Ian R. B. Reeves[1,2,3], Andrew D. Ashton[2], Erika E. Lentz[3], Christopher R. Sherwood[3], Davina L. Passeri[1], Sara L. Zeigler[1]

[1]U.S. Geological Survey, St. Petersburg Coastal and Marine Science Center, St. Petersburg, FL 33701, U.S.A.
[2]Geology and Geophysics Department, Woods Hole Oceanographic Institution, Woods Hole, MA 02543, U.S.A.
[3]U.S. Geological Survey, Woods Hole Coastal and Marine Science Center, Woods Hole, MA 02543, U.S.A.

*Correspondence to*: Ian R. B. Reeves (ireeves@usgs.gov)

**Abstract.** Models of coastal barrier geomorphic and ecologic change are valuable tools for understanding and predicting when, where, and how barriers evolve and transition between ecogeomorphic states. Few existing models of barrier systems are designed to operate over spatiotemporal scales congruous with effective management practices (i.e., decades/kilometers, referred to herein as "mesoscales"), incorporate important ecogeomorphic feedbacks, and provide probabilistic projections of future change. Here, we present a new numerical model designed to address these gaps by explicitly yet efficiently simulating coupled aeolian, marine, vegetation, and shoreline components of barrier evolution over spatiotemporal scales relevant to management. The Mesoscale Explicit Ecogeomorphic Barrier model (MEEB) simulates subaerial ecomorphologic change of undeveloped barrier systems over kilometers and decades using meter-scale spatial resolution and weekly time step. MEEB applies simplified parameterizations to represent and couple key ecogeomorphic processes: dune growth, vegetation expansion and mortality, beach and foredune erosion, barrier overwash, and shoreline and shoreface change. The model is parameterized and calibrated with observed elevation, vegetation, and water level data for a case study site of North Core Banks, NC, USA; simulated ecogeomorphic change in model hindcasts agrees well with observations, demonstrating both favorable skill scores and qualitatively correct behavior. We also describe an additional model framework for producing probabilistic projections that account for uncertainties related to future forcing conditions and intrinsic stochastic dynamics and demonstrate the probabilistic framework's utility with example forecast simulations. As a mesoscale model, MEEB is designed to investigate questions about future barrier ecogeomorphic change of moderate complexity, offering semi-qualitative predictions and semi-quantitative explanations. For example, MEEB can be used to investigate how climate-induced shifts in ecological composition may alter the likelihood of morphologic impacts or to generate probabilistic projections of ecogeomorphic state change.

## 1 Introduction

Coastal barrier environments are of critical economic, ecologic, and cultural importance, but, as low-lying collections of mobile sediment, are constantly evolving under drivers of both chronic and event-based change. In the face of rising





atmospheric temperatures, projected accelerated relative sea-level rise (RSLR; Sweet et al., 2022), and anticipated changes in tropical storm activity (e.g., Knutson et al., 2020), future barrier evolution remains uncertain. This uncertainty is further complicated by transformations in ecological assemblages related to global climate warming, which have become increasingly apparent within barrier systems in recent decades (e.g., Goldstein et al., 2018; Zinnert et al., 2016) and have the potential to

fundamentally alter the morphology and behavior of coastal barriers (e.g., Reeves et al., 2022; Zinnert et al., 2019). An understanding of when, where, and how ecogeomorphic change in barrier systems is most likely to occur remains of paramount importance to coastal communities looking to prepare for and adapt to future change. Furthermore, the ability to capture future changes to barrier systems can help inform broader understanding of both the functional transformations we may anticipate across the broader coastal landscape, as well as whether protections coastal barriers provide to mainland settings will persist

in the future.

Numerical models capable of simulating across a range of scenarios and conditions offer possibly the best opportunity to predict and understand coastal change and behavior. Prediction of historically unprecedented behavior will be important given uncertainties in future forcing conditions (e.g., RSLR, storm intensity) coupled with inherently complex nonlinear interactions (e.g., feedbacks, multistability) and stochasticity (e.g., storm occurrence, seed dispersal). Numerical models can

inform active planning and management strategies for "undeveloped" barrier systems (i.e., those without sustained residential and/or commercial infrastructure and activity) that are typically intended to preserve and protect ecosystems, infrastructure, and natural resources; provide for human use; mitigate hazards; and inform public expectations. Coastal management practices usually consider timescales of several decades into the future – often with regard to milestones of 2050 and 2100 CE defined by climate change science – over multiple kilometers of coastline. Amongst coastal managers and decision-makers, there is

increasing demand for model projections that both explicitly take into account these spatiotemporal horizons and at the same time provide reliable and sufficiently quantitative predictions (French et al., 2016; Martin et al., 2023). Many models of barrier geomorphology and ecology exist as powerful tools for predicting event-based change or understanding fundamental behaviors and processes (Hoagland et al., 2023; Piercy et al., 2023). While these models provide useful insight to planning and decision-making processes, they tend to lack important features and components that are particularly relevant to the typical goals of

management endeavors: management-relevant spatiotemporal scales (decades and kilometers) and resolutions (meters and weeks), feedbacks between key ecologic and geomorphic processes, and the ability to provide probabilistic projections of future change.

Numerical models of barrier evolution can be arranged along a continuum between micro and macro scales (Hoagland et al., 2023; Murray, 2003). What we herein consider microscale models (e.g., XBeach, Roelvink et al., 2009; Delft3D, Lesser

et al., 2004; COAWST, Warner et al., 2008), also commonly referred to as event-based or simulation models, typically simulate coastal change over hours to years and up to hundreds of meters. These models tend to be built upon highly realistic expressions of the underlying physics and incorporate as many system processes as practical while striving to simulate a particular place or set of conditions with as much quantitative accuracy (predictive skill) as possible (Murray, 2003; Sherwood et al., 2022). As such, microscale models typically require relatively large computational resources, observational or experimental data for



model initialization and testing, and careful calibration of important model coefficients (e.g., Windsurf, Itzkin et al., 2022). In contrast, macroscale models (e.g., CoastMorpho2D, Mariotti, 2021; Barrier3D, Reeves et al., 2021; BIT, Masetti et al., 2008; BRIE, Nienhuis and Lorenzo-Trueba, 2019a), often referred to as exploratory or reduced-complexity models, operate over temporal scales of decades to millennia and over spatial scales up to thousands of meters, typically with coarse spatial resolutions ≥ 10 m. Macroscale models simplify systems to focus on essential, emergent processes, often with the goal of

exploring and explaining large-scale behavior (Murray, 2003). As such, larger-scale models tend to use synthesized representations of natural phenomena that average over ecogeomorphic processes and features occurring at much smaller spatiotemporal scales, providing the most direct explanations and likely the most reliable predictions of larger-scale phenomena (Murray, 2007). Macroscale models also tend to use idealized (e.g., LTA14, Lorenzo-Trueba and Ashton, 2014) or equilibrium (e.g., GEOMBEST, Stolper et al., 2005) morphologies meant to represent generalized conditions or behaviors

rather than specific real-world locations.

          There is a dearth, however, of mesoscale models that occupy the continuum between micro- and macroscale endmembers. Few models or model frameworks of coastal barrier environments are designed to simulate over years to decades and hundreds to thousands of meters and with meter-scale spatial resolution. Notable exceptions include DUBEVEG (Keijsers et al., 2016) and the Coastal Dune Model (Durán Vinent & Moore, 2015), which simulate decadal ecologic and geomorphic

evolution of beach and dune environments, and Robson et al. (2024) and ISLAND (Rastetter, 1991), which model decadal vegetation-habitat interactions across a barrier; however, these models lack full representation of the entire barrier system and/or key processes (e.g., overwash) needed for holistic assessments and relevant mesoscale projections of barrier evolution. The polarity of spatiotemporal scales among existing coastal barrier models likely exists because of contrasting goals, assumptions, and modeling techniques of micro- and macroscale models (Murray, 2003); the sheer number of processes that

could be important for driving coastal change (van Maanen et al., 2016); a lack of decadal observational data needed to develop mesoscale parameterizations and evaluate mesoscale models (French et al., 2016; Hoagland et al., 2023); and nascent theory on model up-scaling and down-scaling, with up-scaling approaches (i.e., using microscale models in meso- or macroscale applications) particularly limited by high computational costs and the potential for imperfections in reductionist microscale parameterizations compounding over much larger scales thereby preventing reliable quantitative results (Murray, 2007; French

et al., 2016). However, coastal management practices typically consider timescales on the order of several decades into the future, stemming largely from the well-established climate and sea-level rise horizons of 2050 and 2100 CE. Additionally, continuous, spatially explicit coverage of an area of interest in both the cross-shore and alongshore dimensions, as well as weekly to annual (temporal) and meter to decameter (spatial) resolutions, are needed to inform comprehensive management scenarios (e.g., Martin et al., 2023). As such, barrier models operating at mesoscales are perhaps most promising for addressing

the needs of strategic coastal planning and decision-making (French et al., 2016; van Maanen et al., 2016; Woodroffe and Murray-Wallace, 2012).

          Incorporation of ecological dynamics and their feedbacks with geomorphic processes is also underrepresented in models of coastal barrier evolution (Hoagland et al., 2023; Piercy et al., 2023). Bidirectional physical-ecological feedbacks





within coastal barrier environments are especially relevant at mesoscales, where vital ecogeomorphic behaviors tend to emerge.

At much smaller scales (i.e., years and $10^1$ meters, or smaller), spatiotemporal change in the size, location, or type of herbaceous and woody vegetation found in coastal barrier environments tends to be prohibitively small for generating dynamic ecogeomorphic interactions and feedbacks; at much larger spatiotemporal scales (i.e., centuries or millennia and $10^1$ kilometers, or larger), the influence of vegetation on geomorphic processes tends to become increasingly difficult to recognize (Larsen et al., 2021). It is well documented that ecogeomorphic interactions and feedbacks are fundamental to coastal barrier

systems, from the stimulation of dune-growing grasses in response to sand deposition (e.g., Zarnetske et al., 2012) to woody shrubs obstructing overwash flow (Reeves et al., 2022; Zinnert et al., 2019), yet these interactions are often neglected in barrier models. Modeling approaches that implicitly incorporate the effects of vegetation through static parameterizations, such as landcover roughness coefficients (e.g., Passeri et al., 2018), are typically not appropriate for addressing questions related to meso- or macroscale evolution where the configuration of vegetation is expected to change across space and time. However,

some models of barrier systems have begun to include physical-biological feedbacks by explicitly simulating the spatiotemporal variation of vegetation communities across coastal barrier landscapes and their couplings with physical processes (e.g., ISLAND, Rastetter, 1991; Barrier3D, Reeves et al., 2022; DUBEVEG, Keijsers et al., 2016; Coastal Dune Model, Durán Vinent & Moore, 2015; GEOMBEST++Seagrass, Reeves et al., 2020). Given the sensitivity of coastal ecosystems to changes in climatic forcings and potential ecological transformations arising from global climate change (e.g.,

Goldstein et al., 2018; Jackson et al., 2019; Zinnert et al., 2016; Lucas and Carter, 2010), ecogeomorphic interactions are likely to play an increasingly prominent role over the next century. Including dynamic ecogeomorphic couplings within coastal barrier models therefore improves the performance of mesoscale projections and confidence in their findings.

        Most barrier-evolution models provide only deterministic projections, despite often considerable uncertainty in projections of macroscale drivers of coastal change (e.g., sea-level rise, storminess, atmospheric temperature) and the inherent

randomness of natural phenomena (e.g., storm timing). Significant uncertainties also arise from future human activities and decision-making (e.g., McNamara and Lazarus, 2018), but coupled human-natural considerations are beyond the present scope of this model. While deterministic models inherit the uncertainties of model input forcings, model approaches that account for the probabilistic nature of the drivers of barrier evolution (e.g., Lentz et al., 2016; Bamunawala et al., 2021; Wainwright et al., 2015) can potentially provide a more holistic assessment of future change to better inform management activities (e.g., van

der Lugt, 2019).

        Here we present the Mesoscale Explicit Ecogeomorphic Barrier model (MEEB), which simulates the spatially explicit ecogeomorphic change of undeveloped barrier systems over several decades and kilometers. MEEB tackles the separation of scales among pre-existing barrier models by explicitly yet efficiently simulating aeolian, marine, shoreline, and vegetation components of a coastal barrier segment. Additionally, the model incorporates uncertainties related to future drivers and the

inherent stochasticity of natural processes to produce probabilistic projections of future change across space and time. The goal of MEEB is to balance management needs for spatially explicit, quantitative predictions with mesoscale (multi-decadal and multi-kilometer) projections and related uncertainties while accounting for real feedbacks between ecosystems,



geomorphology, and hydrodynamic processes. To accomplish this, we use a hybrid model architecture that incorporates certain model parameterizations of higher mechanistic complexity, only as far as to produce mesoscale behaviors anticipated to be

important, with more simplified, larger-scale parameterizations that most reliably capture the collective effects of many processes happening at much smaller scales (Thornhilll et al., 2015; French et al., 2016; Murray, 2007). As an important part of our hybrid approach, we also explicitly and robustly test and calibrate the relatively simple model parametrizations with observational data. MEEB is not suitable for predicting subtle shifts in elevation or vegetation, nor for explaining the reconfiguration of a landscape or its behaviors. Rather, MEEB is designed to answer questions of moderate complexity

regarding when, where, and how ecogeomorphic change is likely to occur, with correspondingly moderate levels of both predictive (quantitative) and explanatory (qualitative) power. In the sections that follow, we provide a description of the model processes and parameterizations; detail data integration and calibration procedures; and assess and discuss model performance, parameter sensitivities, and appropriate use.

## 2 MEEB

The Mesoscale Explicit Ecogeomorphic Barrier model (MEEB; Reeves, 2024) resolves cross-shore and alongshore variations in topography and ecology to simulate the ecogeomorphic evolution of an undeveloped barrier or barrier segment (Fig. 1). MEEB operates with a planform grid resolution on the order of meters and 0.02-y (7.3-d) timesteps and can run for years to decades over $10^0$-$10^1$ km of barrier shoreline. We use a grid resolution of 1 m as herbaceous species in barrier ecosystems typically grow in clusters on the order of 1 m$^2$ in size, though the model produces qualitatively similar output with

resolutions up to 2 m (resolutions outside this range have not been tested). Spatial resolution of the model is dictated in large part by the typical size of the vegetation growth pattern as the vegetation imposes a characteristic length scale on the resultant aeolian morphology (Nield & Baas, 2008a). Therefore, we do not suggest adopting a grid resolution outside the 1 to 2 m range. The model is written in Python, and the typical runtime for a 10-year simulation of a 1-km-long barrier segment with 1-m grid resolution is approximately 40 to 80 min, or approximately 10 to 20 min with a grid resolution of 2 m. The relative balance

between spatiotemporal resolution, spatiotemporal extent, and efficiency makes the model ideal for studying mesoscale barrier evolution.

The model tracks changes in elevation and vegetation density and type through space and time across separate elevation and vegetation domains. Ecogeomorphic evolution in MEEB is limited to above the mean high water (MHW) elevation; intertidal and subaqueous environments are therefore not explicitly modeled. The MHW elevation changes each

model iteration according to the RSLR rate, which is constant through time (given that RSLR projections up to 2050 can be closely approximated as linear). A groundwater lens is modeled as a function of the subaerial topography and may intersect depressions in the land surface as ponds. Due to complexities in modeling inlet dynamics, MEEB is not capable of simulating portions of a barrier chain with or directly influenced by active tidal inlets or shoals from recently abandoned inlets. Additionally, MEEB assumes the barrier system is composed entirely of unconsolidated sand. MEEB is initialized with






**Figure 1: MEEB model configuration. (a) Example model elevation domain, annotated with the foredune crestline (red) and ocean shoreline (purple), and (b) corresponding vegetation density domain. (c) Schematic diagram of sand slab transport in the Aeolian component of MEEB, wherein slabs are stochastically entrained, transported downwind, and either deposited or transported**
**further; each of these processes is affected by vegetation, shadow zones, and the water table. (d) Probabilities of slab erosion and deposition as a function of vegetation density. (e) Annual vegetation growth for burial-tolerant and burial-intolerant species types as a function of the annual net sedimentation balance.**





elevation, vegetation cover, and high-water event climatology data, which we discuss in Sect. 3 below. Tables 1 and 2 list all
model parameters and dependent variables, respectively, as well as their units and values.

## 2.1 Model Framework and Time-stepping

Four components comprise the MEEB framework: aeolian, marine, shoreline, and vegetation. These four components operate in succession within a model timestep but act at different timescales, so not all components are involved in each model iteration (Fig. 2). A model year begins with the Aeolian component, which occurs every model timestep ($\Delta t_a = 0.02$ y or ~7.3
d). The Aeolian component determines the entrainment, transport, and deposition of sand across the barrier surface resulting from wind and dependent on the vegetation cover and topography. This Aeolian process repeats itself twice, updating the elevation domain each iteration, and is then followed by the Marine component, which occurs every second model timestep ($\Delta t_m = 0.04$ y or ~14.6 d) to correspond to a spring-neap tidal cycle and because storm systems can potentially last more than 7 days. The Marine component determines how sediment is transported across the beach, dune, and barrier interior from swash,
collision, and overwash processes during a high-water event (HWE), defined as an event in which the total water level exceeds MHW. These HWE-induced processes are influenced by the topography, vegetation cover, and HWE water elevation and duration. In addition to updating the elevation domain, the Marine component also updates the vegetation domain by converting previously vegetated cells to bare wherever inundated. The Shoreline component, which also occurs every second model timestep ($\Delta t_s = 0.04$ y or ~14.6 d), directly follows the Marine component. The Shoreline component determines the position
of the MHW ocean shoreline according to RSLR and cross-shore and alongshore sediment transport, and adjusts the shoreline by adding (accretion) or removing (erosion) elevation. This sequence of two Aeolian iterations, one Marine iteration, and one Shoreline iteration repeats for a total of 25 times in the model year. Thereafter, the full model year is completed with execution of the Vegetation component, which occurs every 50 model timesteps ($\Delta t_v = 1.0$ y or 365 d). The Vegetation component determines the expansion of plants into previously bare cells and changes in vegetation density (growth or decay), the latter of
which is dependent upon the net erosion/deposition over the course of the preceding year. The cycle then restarts with the updated elevation and vegetation domains for the next model year (Fig. 2).

## 2.2 Aeolian

MEEB uses a cellular model of aeolian morphologic development in which slabs of sand are probabilistically entrained, transported, and deposited based on a set of rules that capture the effects of real-world aeolian processes. Our
Aeolian component stems from the aeolian side of the DUBEVEG model (Dune, BEach, and VEGetation; Keijsers et al., 2016), which itself builds upon earlier cellular slab-based dune models (Baas, 2002; Baas & Nield, 2007; Werner, 1995). The Aeolian component in MEEB updates the topography 50 times a year (time step $\Delta t_a = 0.02$ y or ~7.3 d).

During each Aeolian iteration, every cell in the domain is polled once for entrainment based on a probability of erosion ($P_e$) ranging from zero to one, as discussed below. If entrainment in a cell is probabilistically determined to occur, a
slab of sand with a fixed height ($H_s$) is removed from the entrainment site and transported downwind according to the saltation





**Table 1: MEEB parameters and their definitions.**

| Parameter | Units | Value | Calibration and Sensitivity Analysis Range | Description |
|---|---|---|---|---|
| $\Delta t_a$ | y | 0.02 | | Aeolian iteration duration |
| $\Delta t_m$ | y | 0.04 | | Marine iteration duration |
| $\Delta t_s$ | y | 0.04 | | Shoreline iteration duration |
| $\Delta t_v$ | y | 1 | | Vegetation iteration duration |
| $RSLR$ | m yr$^{-1}$ | 0.004 (hindcasts), 0.0068 to 0.0124 (projections) | | Relative sea-level rise |
| $MHW$ | m NAVD88 | 0.39 | | Mean high water |
| **Aeolian** | | | | |
| $P_{e,0}$ | - | 0.10 | 0.02 to 0.5 | Maximum probability of erosion in the complete absence of vegetation cover ($\rho = 0$) |
| $P_{d,0}$ | - | 0.22 | 0.02 to 0.5 | Probability of deposition in the complete absence of vegetation cover ($\rho = 0$) |
| $P_{d,1}$ | - | 0.54 | 0.05 to 0.5 | Probability of deposition with full effective vegetation cover ($\rho = 1$) |
| $\eta$ | deg | 12 | 8 to 18 | Shadow angle |
| $\rho_{q0}$ | - | 0.10 | 0.05 to 0.55 | Vegetation density at which entrainment of sand becomes effectively negligible |
| $\rho_v$ | - | 0.35 | 0.05 to 0.4 | Threshold vegetation density at which cells are considered vegetated |
| $\theta_{r,u}$ | deg | 20 | 15 to 30 | Angle of repose for unvegetated cells ($\rho < \rho_v$) |
| $\theta_{r,v}$ | deg | 30 | 20 to 40 | Angle of repose for vegetated cells ($\rho \geq \rho_v$) |
| $P_{WD}$ | - | (0.81, 0.04, 0.06, 0.09) | 0.5 to 1 (wind axis ratio); 0.5 to 1 (onshore wind ratio); 0 to 1 (down-shore wind ratio) | Wind direction probability (onshore, alongshore down, offshore, alongshore up) |
| $H_s$ | m | 0.02 | | Aeolian slab height |
| $L_s$ | m | 5 | | Saltation length |
| $D_{gw}$ | - | 0.4 | | Proportion of the smoothed topography above MHW for determining elevation of the freshwater lens |
| **Marine** | | | | |
| $\beta_{eq}$ | - | 0.022 | 0.01 to 0.04 | Equilibrium beach slope |
| $T_e$ | - | 1.48 | 1.0 to 3.0 | Erosive timescale calibration coefficient for sediment flux seaward of the foredune crest |
| $R_d$ | m$^3$ h$^{-1}$ | 249 | 50 to 280 | Parameter representing infiltration and drag of overwash flow |
| $S_{Qlim}$ | - | 1.5 | 0.5 to 2.0 | Maximum slope water can flow uphill |
| $K_{ow}$ | - | 0.0001684 | 0.00005 to 0.01 | Overwash sediment transport coefficient |
| $C_s$ | - | 0.0283 | 0.01 to 0.04 | Constant representing momentum of the overwash flow |



| $m$ | - | 1.04 | 1.01 to 1.12 | Constant for nonlinear relationship between sediment flux and discharge |
|---|---|---|---|---|
| $g$ | m s$^{-2}$ | 9.81 | | Gravitational acceleration |
| $n$ | - | 0.5 | | Constant for flow routing |
| $C_{bb}$ | - | 0.7 | | Coefficient for exponential decay of sediment load entering subaqueous back-barrier environment |
| $D_{bb}$ | m | 1.5 | | Equilibrium depth of back-barrier basin |
| $t_{s\_l}$ | hr | 0.02 | | Time-substep for computing an hourly iteration of Marine HWE morphological change landward of the dune crest |
| $t_{s\_s}$ | hr | 0.04 | | Time-substep for computing an hourly iteration of Marine HWE morphological change seaward of the dune crest |
| **Vegetation** | | | | |
| $\Lambda$ | - | 0.02 (burial-tolerant); 0.05 (burial-intolerant) | 0.02 to 0.4 (burial-tolerant); 0.05 to 0.5 (burial-intolerant) | Vegetation-overwash flow reduction coefficient |
| $P_{germ}$ | - | 0.05 | | Probability of vegetation establishment via germination from seeds or rhizome fragments |
| $P_{lat}$ | - | 0.2 | | Probability of vegetation establishment via lateral expansion from neighboring vegetated cells |
| $V_{z,min}$ | m MHW | 0.25 (burial-tolerant); 0.25 (burial-intolerant) | | Minimum elevation (relative to MHW) for vegetation |
| $V_{x,a-e}$ | m y$^{-1}$ | [-1.5, -0.05, 0.5, 1.5, 2.2] (burial-tolerant); [-1.6, -0.7, 0, 0.2, 2.1] (burial-intolerant) | | X-coordinates of the 5 growth function vertices $a$-$e$ |
| $V_{y,c}$ | y$^{-1}$ | 0.2 (burial-tolerant); 0.05 (burial-intolerant) | | Peak growth of the growth function middle vertex $c$ |
| **Shoreline** | | | | |
| $D_{sf}$ | m | 20.07 | | Shoreface depth |
| $S_{sf,eq}$ | - | 0.00822 | | Equilibrium shoreface slope |
| $k_{sf}$ | m$^3$ m$^{-1}$ y$^{-1}$ | 5926 | | Shoreface flux rate coefficient |
| $y_a$ | m | 25 | | Alongshore length of shoreline sections |
| $H_s$ | m | 0.98 | | Average deepwater wave height |
| $T$ | s | 6.6 | | Average deepwater wave period |
| $R$ | - | 1.65 | | Submerged specific gravity of shoreface sediment |
| $D_{50}$ | m | 2e-4 | | Median grain size of shoreface sediment |
| $e_s$ | - | 0.01 | | Shoreface suspended sediment transport efficiency factor |
| $c_s$ | - | 0.01 | | Shoreface friction factor |
| $k_d$ | m$^{3/5}$ s$^{-6/5}$ | 0.06 | | Shoreline diffusivity constant |



| $a$ | - | 0.6 | | Wave climate asymmetry, i.e., proportion of waves approaching from the left when looking offshore of the regional shoreline trend |
|---|---|---|---|---|
| $h$ | - | 0.39 | | Proportion of high-angle waves, i.e., waves with an approach angle greater than 45º |

**Table 2: MEEB dependent variables and their definitions.**

| Variable | Units | Description |
|---|---|---|
| **Aeolian** | | |
| $P_e$ | - | Probability of erosion |
| $P_d$ | - | Probability of deposition |
| $q_{a,max}$ | m³ m⁻¹ $\Delta t_a^{-1}$ | Maximum potential aeolian transport flux |
| $\rho$ | - | Vegetation density |
| **Marine** | | |
| $Z_{twl}$ | m NAVD88 | Total water level elevation of high-water event |
| $Z_x$ | m NAVD88 | Elevation at cross-shore location $x$ |
| $x_D$ | m | Cross-shore location of the foredune crest |
| $q_x$ | m³ s⁻¹ | Flux of sediment at cross-shore location $x$ seaward of the foredune crest |
| $\beta_x$ | - | Local slope |
| $Q_{dc}$ | m³ h⁻¹ | Discharge over the foredune crest |
| $Z_D$ | m NAVD88 | Foredune crest elevation |
| $U$ | m h⁻¹ | Velocity of water at the dune crest |
| $Q_0$ | m³ h⁻¹ | Overwash discharge at distributing cell |
| $Q_i$ | m³ h⁻¹ | Overwash discharge at receiving cell $i$ |
| $S_i$ | - | Local directional slope from distributing cell to receiving cell $i$ |
| $q_{si}$ | m³ h⁻¹ | Overwash sediment flux from distributing cell to receiving cell $i$ |
| **Vegetation** | | |
| $Q_{i,eff}$ | m³ h⁻¹ | Effective discharge leaving vegetated cells |
| **Shoreline** | | |
| $x_s$ | m | Cross-shore position of the ocean shoreline |
| $x_t$ | m | Cross-shore position of the shoreface toe |
| $S_{sf}$ | - | Slope of the active shoreface |





| $q_{sf}$ | m$^3$ m$^{-1}$ $\Delta t_s^{-1}$ | Shoreface sediment flux |
|---|---|---|
| $q_{ow}$ | m$^3$ m$^{-1}$ $\Delta t_s^{-1}$ | Cumulative volume of overwash deposition deposited on and behind the barrier interior for 1 Shoreline iteration |
| $q_{bd}$ | m$^3$ m$^{-1}$ $\Delta t_s^{-1}$ | Cumulative volume of sediment imported from or exported to the beach and dune system during HWEs for 1 Shoreline iteration |
| $w_s$ | m s$^{-1}$ | Settling velocity of shoreface sediment |
| $z_0$ | m | Breaking wave depth |
| $D_j$ | m$^2$ s$^{-1}$ | Shoreline diffusivity |
| $\theta$ | - | Shoreline angle relative to the regional shoreline trend |
| $\phi_0$ | - | Wave angle relative to the regional shoreline trend |
| $\Psi$ | - | Dependence of the diffusivity to the wave angle |
| $E$ | - | Normalized angular distribution of wave energy |





Figure 2: Flow diagram and schematic illustrations of model process domains across one model year in MEEB.





length ($L_s$). A probability of deposition ($P_d$) at the receiving cell determines whether the slab will deposit or be transported downwind again (Fig. 1c). As a proxy for weekly time-varying wind speeds, we stochastically vary $L_s$ each Aeolian iteration by drawing from a simple uniform distribution centered around a mean (here, $5 \pm 2$ m). As a proxy for the longer-term (annual- or decadal-scale) average strength of the wind climate, the maximum potential aeolian transport volume flux ($q_{a,max}$) can be calculated following Nield and Baas (2008a) as:


$$q_{a,max} = H_s L_s \frac{P_e}{P_d} \frac{1}{\Delta t_a} \,. \tag{1}$$

Changing the values $P_e$ and $P_d$ therefore captures the effects of stronger or weaker wind climates. Wind direction – onshore, alongshore down, offshore, or alongshore up across the gridded domain – is chosen randomly for each Aeolian iteration, with

the probability of each of the 4 directions ($P_{WD}$) summing to 1. The collective effects of oblique winds are roughly captured with asymmetric multidirectional transport directions (cf. Nield & Baas, 2008a). At the end of each Aeolian iteration, angles of repose for bare and vegetated cells are maintained by avalanching slabs in the direction of steepest descent. The Aeolian component uses open boundary conditions wherein slabs of sediment can be transported out of the lateral edges of the model domain (sediment can be imported into the domain within the Marine component of the model, as described below).

Probabilities of erosion and deposition vary across the landscape as function of ecological and physical factors. Wind shadow zones, wherein $P_e = 0$ and $P_d = 1$, extend from the lee side of topographic peaks as determined by the shadow angle $\eta$ (Fig. 1c). Where elevation is below MHW or the elevation of a groundwater lens, $P_e = 0$ and $P_d = 1$. The groundwater surface is determined as a proportion of the topographic surface height above MHW (Fig. 1c; Galiforni Silva et al., 2018) that has been smoothed by a Gaussian filter (with a standard deviation for the Gaussian kernel, $\sigma$, of 12 m), and groundwater can

intersect topographic depressions as surface ponds. Additionally, the presence of vegetation cover reduces the probability that a slab will be eroded and increases the probability that a slab will be deposited in proportion to the vegetation density ($\rho$) of each cell (Fig. 1d). $P_e$ decreases linearly from its maximum ($P_{e,0}$) when $\rho = 0$ to 0 when $\rho = \rho_{q0}$, the vegetation density at which entrainment of sand becomes effectively negligible. Where vegetation is present in cells between the entrainment site and receiving site along the transport path, entrained slabs are polled for deposition at each intermediary vegetated cell if $\rho$ is

greater than a threshold value $\rho_v$ (Teixeira et al., 2023). To account for the effects of vegetated cells on the local wind field of neighboring unvegetated cells, the Aeolian component of MEEB uses a copy of the current vegetation density domain that has been lightly smoothed with a Gaussian filter ($\sigma = 3$ m).

## 2.3 Marine

High-water events (HWEs) occur intermittently in MEEB, causing changes to the subaerial morphology and ecology.

We define HWEs as all events in which the total water level exceeds MHW. As discussed below, MEEB uses separate – but coupled – model formulations seaward and landward of the dynamic foredune crestline (described in Sect. 2.3.1 below) to





simulate ecogeomorphic change from HWEs. The model can run hindcast simulations using time series of observed HWEs as input or run forecast simulations with a stochastic HWE environment developed from the time series of observed HWEs. Our methodology for the observational HWE time series and stochastic HWE environment is described in Sect. 3.3, below.

HWEs in MEEB are described by a total water level (TWL, the representative highest elevation of the landward margin of runup), which can vary alongshore according to the local beach slope, and a duration (in hours). Every $25^{-1}$ y ($\Delta t_m$ = 0.04 y or ~14.6 d), MEEB determines whether a HWE occurs depending on the observed time series (for hindcasts) or a probability of occurrence dependent on the time of year (for forecasts). In the stochastic HWE environment, the conditions of each event are chosen randomly from a list of synthetic HWEs; the probability of occurrence and average intensity (TWL and

duration) of the list of synthetic HWEs, however, remain constant over the course of each simulation. If no HWE is determined to occur for a Marine iteration, no marine processes take place (i.e., the landscape remains unaltered) and MEEB proceeds directly to the Shoreline component of the model. For simplicity, and because our identification of HWEs tend to lump multiple tidal cycles of an event together, MEEB allows a maximum of one HWE to occur during each 0.04-y (14.6-d) interval.

### 2.3.1 Dune crest location

MEEB uses separate formulations for HWE-driven morphologic change landward and seaward of the foredune crest, therefore requiring the alongshore-continuous location of the foredune ridge. MEEB identifies the cross-shore locations of the foredune crest ($x_D$) for every $dy$ alongshore – the foredune crestline – using a multi-step process that considers the general trend of the foredune crest location to identify gaps in the crestline where the dune would be most likely to (re)form. First, the algorithm finds the cross-shore locations of the elevation maximum for an elevation domain that has been smoothed in the

alongshore dimension using a large-window (150-m) moving average. This maximum elevation crestline is smoothed again with a Savitzky-Golay filter (window length = 75 m), resulting in a demarcation that gives the broad, general trend of where the foredune crest is or would tend to be (in the case of gaps in the foredune) located within the barrier domain. Next, using the original non-smoothed elevation domain, the algorithm finds the location of the dune-crest peaks within a 25-m buffer of this broad crest trendline. Dune-crest peaks are selected as the most-seaward peak of a profile with a minimum backshore drop

of 0.6 m (Itzkin et al., 2020; Mull and Ruggiero, 2014). If no peak is found within a profile, the algorithm selects the location of the maximum elevation within the 25-m buffer on either side of the broad crest trendline and the location alongshore is considered a gap in the foredune crestline. Lastly, a Savitzky-Golay filter is applied again with a smaller window length (11 m) to produce the continuous foredune crestline.

### 2.3.2 Ocean shoreline to foredune crest

MEEB uses an equation for cross-shore net sediment transport between the surf-swash boundary and the foredune crest to simulate HWE processes seaward of the foredune crest. Based on Duran & Moore (2015) and, by extension, Larson et al. (2004a), the flux of sediment at cross-shore location $x$, $q_x$, is equal to





$$q_x = (\beta_{eq} - \beta_x)(Z_{twl} - Z_x)^2 T_e \tag{2}$$


with $\beta_{eq}$ the representative equilibrium slope of the beach, $\beta_x$ the local slope at the cross-shore location $x$, $Z_{twl}$ the TWL elevation, $Z_x$ the elevation at the cross-shore location $x$, and $T_e$ a calibration coefficient for the erosive timescale (see Tables 1 and 2 for a list of model parameters and dependent variables, their units, and values used herein). Sediment transport for each HWE iteration is calculated from the ocean shoreline up to either the first cross-shore location at which $Z_x$ exceeds $Z_{twl}$,

beyond which $q_x = 0$, or the crestline, beyond which sediment flux follows the overwash flow routing scheme described in Sect. 2.3.3. Transport depends on deviation from the equilibrium beach slope, as these local interactions nudge the beach volume towards a linear equilibrium configuration over time. As Eq. (2) calculates only cross-shore sediment fluxes, sediment flux in the alongshore dimension is not incorporated. Change in elevation ($\Delta Z_x / \Delta t$) is calculated as the divergence of $q_x$ in the cross-shore dimension:


$$\frac{\Delta Z_x}{\Delta t} = \frac{-\Delta q_x}{\Delta x} . \tag{3}$$

### 2.3.3 Foredune crest to back-barrier shoreline

To simulate HWE processes landward of the foredune crest, MEEB utilizes a version of the overwash flow routing formulation from Barrier3D (Reeves et al., 2021). Water introduced at overtopped dune cells is transported landward cell-by-

cell, carrying sediment with it. After water and sediment have been routed across the barrier interior, the elevation of the barrier interior is updated according to the sediment flux into and out of each cell. This process occurs iteratively with hourly time steps for the duration of the HWE.

Water discharge is introduced at each overtopped dune crest cell ($Q_{dc}$) according to


$$Q_{dc} = U(Z_{twl} - Z_D) \tag{4}$$

with $Z_D$ the dune-crest elevation, $g$ gravitational acceleration, and $U$ the velocity of the water at the dune crest (Larson et al., 2004b):


$$U = \sqrt{2g(Z_{twl} - Z_D)} . \tag{5}$$

where $g = 9.81$ m s$^{-2}$ is gravitational acceleration. Water is distributed to the three neighboring cells in the next landward row of the domain in proportion to the local slope. If any of the slopes to the 3 landward neighbors are positive (i.e., downhill), the neighbor with the steepest downhill slope will receive the most water:






$$Q_i = \frac{(Q_0 - R_d)S_i^n}{\sum S_i^n} \tag{6}$$

where $Q_0$ is the discharge at the distributing cell, $Q_i$ is the discharge and $S_i$ the directional slope from the distributing cell to the landward neighbor $i$, $R_d$ is a parameter to represent infiltration and drag, and $n$ is a constant equal to 0.5. If all of the slopes

to the three landward neighbors are negative (i.e., uphill), the neighbor with the least steep uphill slope receives the most discharge, and the total discharge from $Q_0$ to $Q_i$ is reduced linearly with increasing uphill steepness to the extent of the uphill slope limit ($S_{Qlim}$):

$$Q_i = \begin{cases} \frac{(Q_0 - R_d)|S_i|^{-n}}{\sum |S_i|^{-n}}\left(1 - \frac{|S_i|}{S_{Qlim}}\right), & S_i < S_{Qlim} \\ 0, & S_i \geq S_{Qlim} \end{cases}. \tag{7}$$


Neighboring cells with slopes greater than $S_{Qlim}$ will therefore receive no discharge.

The volume of sediment transported from the distributing cell to landward neighbor $i$, $q_{si}$, depends on the discharge and local slope (i.e., the stream power index, $QS$; Murray and Paola, 1997):


$$q_{si} = K_{ow}[Q_i(S_i + C_s)]^m \tag{8}$$

where $K_{ow}$ is a dimensional sediment-transport coefficient, $C_s$ is a non-dimensional constant representing flow momentum (on the order of several times the average slope of the barrier interior), and $m$ is a constant greater than 1. In contrast to the formulation in Barrier3D, MEEB allows upslope sediment transport and does not distinguish between different overwash

regimes (i.e., run-up versus inundation) when determining parameter values and transport equations.

Where overwash reaches the back-barrier shoreline, the sediment load into the subaqueous back-barrier environment is distributed in an exponentially decaying fashion with the landward neighbor with the most discharge receiving the most sediment:


$$q_{si} = \frac{(q_{s0}C_{bb})Q_i}{\sum Q_i} \tag{9}$$

with $q_{s0}$ the flux of sediment transported into the distributing cell, and $C_{bb}$ the decay coefficient set to a value that produces steeply dipping delta-like foreset deposits typically observed when overwash flows into standing bodies of waters (Schwartz, 1982). MEEB assumes that the bottom of the back-barrier bay is flat and that depositional and erosional processes can maintain





a constant equilibrium back-barrier depth ($D_{bb}$) relative to MHW over the course of the simulation (Marani et al., 2007). This assumption excludes the potential for back-barrier depth to change over space and time, for example via complex tidal bathymetry or the expansion of subtidal and/or intertidal landforms, but these dynamics are outside the present scope of the model.

### 2.3.4 Temporal discretization

To avoid instabilities, we compute each hourly storm iteration with a finer substep, smaller than some upper bound, for both the landward ($t_{s\_l}$) and seaward ($t_{s\_s}$) formulations of the Marine component. Smaller substeps maximize model skill (up to a point), while larger substeps (still small enough to avoid instabilities) maximize model speed. The size of substeps chosen for simulations herein ($t_{s\_l} = 0.02$ h and $t_{s\_s} = 0.04$ h) tend more towards model skill than efficiency, though significant improvements in model speed are likely possible with larger substeps while sacrificing comparatively little model skill.

### 2.3.5 Coupling marine formulations across the foredune crest boundary

Although MEEB uses separate formulations for HWE-driven morphologic change landward and seaward of the foredune crest, the formulations are coupled to produce a smooth transition across this boundary. This coupling is accomplished by sequentially exchanging sediment fluxes and elevations at the foredune crest boundary each hourly timestep as the HWE progresses:

1) The current location of the foredune crestline is determined (Sect. 2.2.1).

    2) Morphologic change landward of the foredune crest occurs (Sect. 2.2.3).

    3) Morphologic change seaward of the foredune crest occurs (Sect. 2.2.2).

    4) Sediment fluxes out of and into the dune crest cells (from steps 2 and 3, respectively) determine the change in elevation at the dune crest boundary.

5) The next hour of the HWE event begins with the updated elevation domain.

### 2.4 Shoreline

In MEEB, change in the cross-shore position of the ocean shoreline ($\Delta x_s$) is the sum of cross-shore ($\Delta x_{s,c}$) and alongshore ($\Delta x_{s,a}$) shoreline change components:

$$\frac{\Delta x_s}{\Delta t_s} = \frac{\Delta x_{s,c}}{\Delta t_s} + \frac{\Delta x_{s,a}}{\Delta t_s} \; . \tag{10}$$

Ocean shoreline change is applied following every Marine iteration ($\Delta t_s = 0.04$ y or ~14.6 d) to sections of the shoreline with a predetermined alongshore length $\Delta y_a$ (typically 25 m), and the shoreline position within each alongshore section is held uniform (Fig. 1a). The initial ocean shoreline position is taken as the intersection of the initial MHW with initial topography





averaged in increments of $\Delta y_a$ alongshore. To implement shoreline erosion (landward change in $x_s$) topographically, previous

beach cells are set to subaqueous elevations that follow a linear shoreface drawn between the new shoreline position at MHW

and the shoreface toe, as described further in Sect. 2.4.1. MEEB implements shoreline accretion (seaward change) by setting

new beach cells to an elevation equal to the average elevation of the previous five most seaward beach cells plus the RSLR for

that shoreline iteration.

**2.4.1 Cross-shore shoreline change**

MEEB follows the equations from the model of Lorenzo-Trueba and Ashton (2014) governing the cross-shore

location of the ocean shoreline ($x_s$) and the shoreface toe ($x_t$), which together determine the slope of the active shoreface

($S_{sf}$):

$$S_{sf} = \frac{D_{sf}}{x_s - x_t} \tag{11}$$

where $D_{sf}$ is the shoreface depth. The shoreface slope is allowed to deviate from its equilibrium slope ($S_{sf,eq}$) in response to

perturbations. When the shoreface steepens past its equilibrium configuration (e.g., as a result of RSLR; Bruun, 1962),

shoreface fluxes are directed offshore; if the shoreface shallows past its equilibrium (which can occur when overwash and

aeolian processes remove sediment from the upper shoreface), shoreface fluxes are directed onshore. Shoreface flux ($q_{sf}$)

therefore depends on the deviations of the shoreface slope from its equilibrium:

$$q_{sf} = k_{sf}\left(S_{sf} - S_{sf,eq}\right) \tag{12}$$

with $k_{sf}$ a dimensional shoreface flux rate coefficient, wherein a larger (smaller) $k_{sf}$ results in faster (slower) adjustment of

the shoreline back towards its equilibrium configuration.

Following Lorenzo-Trueba and Ashton (2014), the cross-shore locations of the ocean shoreline and shoreface toe

evolve as a function of RSLR, the cumulative volume of sediment added to or removed from the upper shoreface, and the net

sediment exchange between the upper and lower shoreface:

$$\Delta x_{s,c} = \frac{2(q_{ow} + q_{bd})}{D_{sf}} - \frac{4q_{sf}}{D_{sf}} \tag{13}$$

$$\Delta x_t = \frac{4q_{sf}}{D_{sf}} + \frac{2(RSLR)}{S_{sf}} \tag{14}$$





where $q_{ow}$ and $q_{bd}$ are the cumulative volumes of sediment per unit of alongshore length imported to or exported from the barrier interior as overwash (landward of the foredune crest) and beach and dune system (seaward of the foredune crest), respectively, during HWEs as determined by comparing pre- and post-event topography. The cross-shore locations of $x_{s,c}$ and $x_t$ will therefore change at relatively similar (dissimilar) rates with a larger (smaller) shoreface flux coefficient $k_{sf}$. Unlike Lorenzo-Trueba and Ashton (2014), we do not extend the effective shoreface above MHW to include the subaerial barrier

height in our formulations for the shoreface mass balance given that MEEB explicitly simulates subaerial morphologic evolution.

MEEB allows the user to optionally estimate $D_{sf}$, $S_{sf,eq}$, and $k_{sf}$ as a function of wave climate and sediment characteristics. Following Nienhuis and Lorenzo-Trueba (2019a),

$$D_{sf} = 0.018 H_s T \sqrt{\frac{g}{R D_{50}}} \tag{15}$$

where $H_s$ and $T$ are the average deepwater wave height and period, $R$ the submerged specific gravity of sediment, and $D_{50}$ the median sediment grain size. The equilibrium shoreface slope can be estimated as

$$S_{sf,eq} = \frac{3 w_s}{4\sqrt{g D_{sf}}} \left( 5 + \frac{3 T^2 g}{4 \pi^2 D_{sf}} \right) \tag{16}$$


where $w_s$ is the settling velocity according to

$$w_s = \frac{R g D_{50}^2}{18 \cdot 10^{-6} + \sqrt{\frac{3}{4} R g D_{50}^3}} . \tag{17}$$


The shoreface response rate can be estimated as

$$k_{sf} = (3600 \cdot 24 \cdot 365) \cdot \frac{e_s c_s g^{\frac{11}{4}} H_s^5 T^{\frac{5}{2}}}{960 R \pi^{\frac{7}{2}} w_s^2} \cdot \frac{\frac{1}{\left(\frac{11}{4}\right) z_0^{\frac{11}{4}}} - \frac{1}{\left(\frac{11}{4}\right) D_T^{\frac{11}{4}}}}{D_T - z_0} \tag{18}$$

with $z_0$ the breaking wave depth ($H_s/0.4$), $e_s$ the suspended-sediment transport efficiency factor, and $c_s$ the friction factor (Nienhuis and Lorenzo-Trueba, 2019a).



### 2.4.2 Alongshore shoreline change

MEEB uses a nonlinear, wave-climate-averaged alongshore diffusion equation for deepwater wave conditions and non-complex coastlines from Nienhuis and Lorenzo-Trueba (2019a), based upon the formulations of Ashton and Murray 430 (2006). Shoreline change from alongshore diffusion is computed using an implicit Crank-Nicholson scheme as:

$$\frac{{x_{s,a}}_j^{t+1} - {x_{s,a}}_j^{t}}{\Delta t_s} = \frac{D_j}{2}\frac{\left({x_{s}}_{j+1}^{t+1} - 2{x_s}_j^{t+1} + {x_s}_{j-1}^{t+1}\right) + \left({x_s}_{j+1}^{t} - 2{x_s}_j^{t} + {x_s}_{j-1}^{t}\right)}{\Delta y_a^{2}} \tag{19}$$

where $t$ and $j$ denote relative time and location of each alongshore section, respectively. $D_j$ is the shoreline diffusivity 435 computed as a function of wave climate:

$$D(\theta) = \frac{k_d}{D_{sf}} H_s^{\frac{12}{5}} T^{\frac{1}{5}}[E(\phi_0)\cdot \Psi(\phi_0 - \theta)] \tag{20}$$

where $\theta$ is the angle of the coastline that varies in space, $\phi_0$ is the offshore wave angle, $k_d$ is an alongshore sediment transport 440 constant set equal to ~0.06 m$^{3/5}$ s$^{-6/5}$ from Nienhuis et al. (2015). $\Psi$ is the dependence of the diffusivity to the wave angle, equal to:

$$\Psi(\phi_0 - \theta) = \cos^{\frac{1}{5}}(\phi_0 - \theta)\left[\cos^2(\phi_0 - \theta) - \frac{6}{5}\sin^2(\phi_0 - \theta)\right] \tag{21}$$

which, averaging over a long-term interannual wave climate, can be convolved with the normalized angular distribution of wave energy $E(\phi_0)$,

$$E(\phi_0) = \begin{cases} a \cdot h, & -\frac{1}{2}\pi < \phi_0 < -\frac{1}{4}\pi \\ a + (1-h), & -\frac{1}{4}\pi < \phi_0 < 0 \\ (1-a)(1-h), & 0 < \phi_0 < \frac{1}{4}\pi \\ (1-a)h, & \frac{1}{4}\pi < \phi_0 < \frac{1}{2}\pi \end{cases} \tag{22}$$

where $a$ is the proportion of waves approaching from the left when looking offshore relative to the regional shoreline trend (i.e., wave climate asymmetry), and $h$ is the proportion of waves with an approach angle greater than 45º (i.e., proportion of high-angle waves), resulting in the long-term, averaged ocean shoreline diffusivity for each section $j$ alongshore. MEEB uses



single representative values of $a$ and $h$ for the entire shoreline (described in Sect. 3.4). For simplicity, MEEB assumes zero-diffusivity boundary conditions, which in effect holds shoreline positions at the edges of the model domain in place within the

alongshore component of shoreline change, $\Delta x_{s,a}$ (the shoreline positions at the edges of the domain, however, can still change within the cross-shore component of shoreline change, $\Delta x_{s,c}$).Therefore, the ocean shoreline will tend towards a linear shape between the two endpoints of the domain, with any perturbations to the shoreline shape (e.g., from overwash) smoothed out over time.

**2.5 Vegetation**

Vegetation dynamics in MEEB follow the vegetation module in DUBEVEG (Keijsers et al., 2016), with the vegetation updating once every year ($\Delta t_v$ = 1 y or 365 d). Each cell in the model domain is described by a vegetation density $\rho$ ranging from 0 (bare) to 1 (fully vegetated); this measure of density is taken as a proxy for the "effectiveness" of the vegetation in its ecogeomorphic interactions (Baas, 2002). In the model, multiple species types with varying ecogeomorphic behaviors can be used concurrently across the domain and may occupy the same cell at any given time. We determine initial

vegetation density using classified landcover datasets from remotely sensed imagery (described in Sect. 3.2.2, below).

The establishment of vegetation into previously bare cells occurs via two mechanisms, either dispersal of seeds and rhizome fragments randomly across the domain or via lateral expansion from neighboring vegetated cells. During each annual iteration, vegetation establishment in previously bare cells is stochastically determined based on the probability of successful germination from seeds or rhizome fragments ($P_{germ}$). For subaqueous cells or cells below a species-specific minimum

elevation relative to MHW ($V_{z,min}$), $P_{germ} = 0$ (MEEB does not simulate the growth and morphodynamics of marsh vegetation). Lateral expansion, or the establishment of vegetation within previously bare cells that neighbor previously vegetated cells (8-cell neighborhood), is stochastically determined based on the probability of lateral expansion ($P_{lat}$).

Growth of established vegetation is modeled as a function of the depositional balance of each vegetated cell (Fig. 1e), capturing key ecogeomorphic couplings of coastal dune systems. The growth functions vary according to the species type.

Here, we model a "burial-tolerant" species type representative of typical dune-building grasses (e.g., *Ammophila* spp., *Uniola paniculata*) that are stimulated by moderate rates of net accretion (Fig. 1e). This positive feedback between plant growth and deposition can give rise to logistic growth behavior of vegetation density in the model (Nield & Baas, 2008b). We also model a "burial-intolerant" species type that is representative of woody vegetation (e.g., *Morella* spp.), which are most productive in the absence of net erosion or accretion and grow more slowly than the burial-tolerant species type (Fig. 1e). Growth functions

are defined by the $x$-coordinates of their five vertices ($V_{x,a-e}$) as well as the peak growth of the middle vertex ($V_{y,c}$; Fig. 1e). Negative growth (i.e., decay) can ultimately result in plant mortality. Vegetation mortality also occurs directly following HWEs wherever vegetated cells are inundated. These present mortality rules are a broad simplification in the model due to complexities in determining vegetation response to overwash disturbances, which depends on species-specific threshold levels of exposure to moisture, salinity, sediment deposition/erosion, and wave action; they also do not distinguish between dead and





buried vegetation, which may allow vegetation in overwashed areas to reestablish more quickly. After an individual plant's density $\rho$ reaches 0, representing mortality, no memory of the plant is preserved; this is a particularly suitable assumption for herbaceous species types which are easily decomposed and/or carried away by wind or water after death, but less so for woody species types that can remain in place and potentially impact barrier ecomorphodynamics years following mortality (Reeves et al., 2022). Nevertheless, plant mortality in the model captures the fundamental response of vegetation to important

environmental stressors.

The presence of vegetation in MEEB influences not only aeolian sediment transport but also overwash sediment transport. Following Barrier3D (Reeves et al., 2022), the effective discharge leaving vegetated cells ($Q_{i,eff}$) is reduced according to the species-specific flow reduction coefficient, $\Lambda$, and vegetation density $\rho$:

$$Q_{i,eff} = Q_i(1 - \Lambda\rho) \tag{23}$$

where $Q_i$ is the calculated discharge leaving neighboring cell $i$ in the absence of vegetation. As such, discharge through vegetated cells is reduced relative to unvegetated cells, with denser vegetation causing more reduction, which in turn tends to cause greater net deposition of sediment within the cell. Future work could upgrade the Vegetation component to incorporate

the effects of seasonality, climate forcing, and more robust representations of environmental filters and plant mortality.

**2.6 Probabilistic framework**

MEEB optionally can be used with a simple probabilistic framework to account for uncertainties related to external and intrinsic stochastic dynamics (Fig. 3). External stochasticity can arise from uncertainty in future forcing conditions, such as RSLR, storm frequency, or mean storm intensity, and is incorporated by running the model across discrete probability

distributions of external drivers, i.e., simulating across a set of values for a particular external forcing variable, each with a specific probability of occurrence that collectively sum to 1. Intrinsic stochasticity within the model simulations arises from inherent randomness of natural phenomena – namely, the probabilistic nature of the storm sequence (timing, water level, duration) and the Aeolian and Vegetation model formulations – and is incorporated with a Monte Carlo method that runs multiple duplicate simulations for each bin of the external forcing probability distribution. The larger the number of duplicate

simulations ($n_P$), the more accurately the sampled distribution represents the theoretical distribution; in our examples below, $n_P = 32$. Uncertainties of multiple external forcing parameters (e.g., RSLR and storm intensity) can be considered together by determining the joint probability of occurrence for scenarios that encompass all possible parameter value combinations, akin to the basic Joint Probability Method (JPM) commonly employed in storm and flood impact analyses (e.g., FEMA, 2016). In the probabilistic framework used herein, RSLR is the only external forcing parameter considered, and we utilize a discrete

probability distribution for RSLR by the year 2050 CE (Fig. 3a) according to the Intergovernmental Panel on Climate Change Shared Socioeconomic Pathway SSP5-8.5 (Fox-Kemper et al., 2021); see Sect. 3.5 for details.





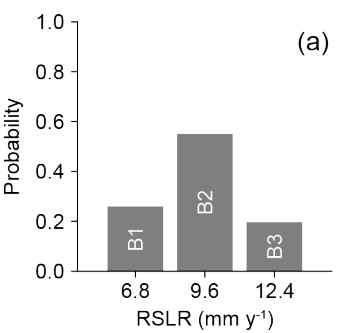

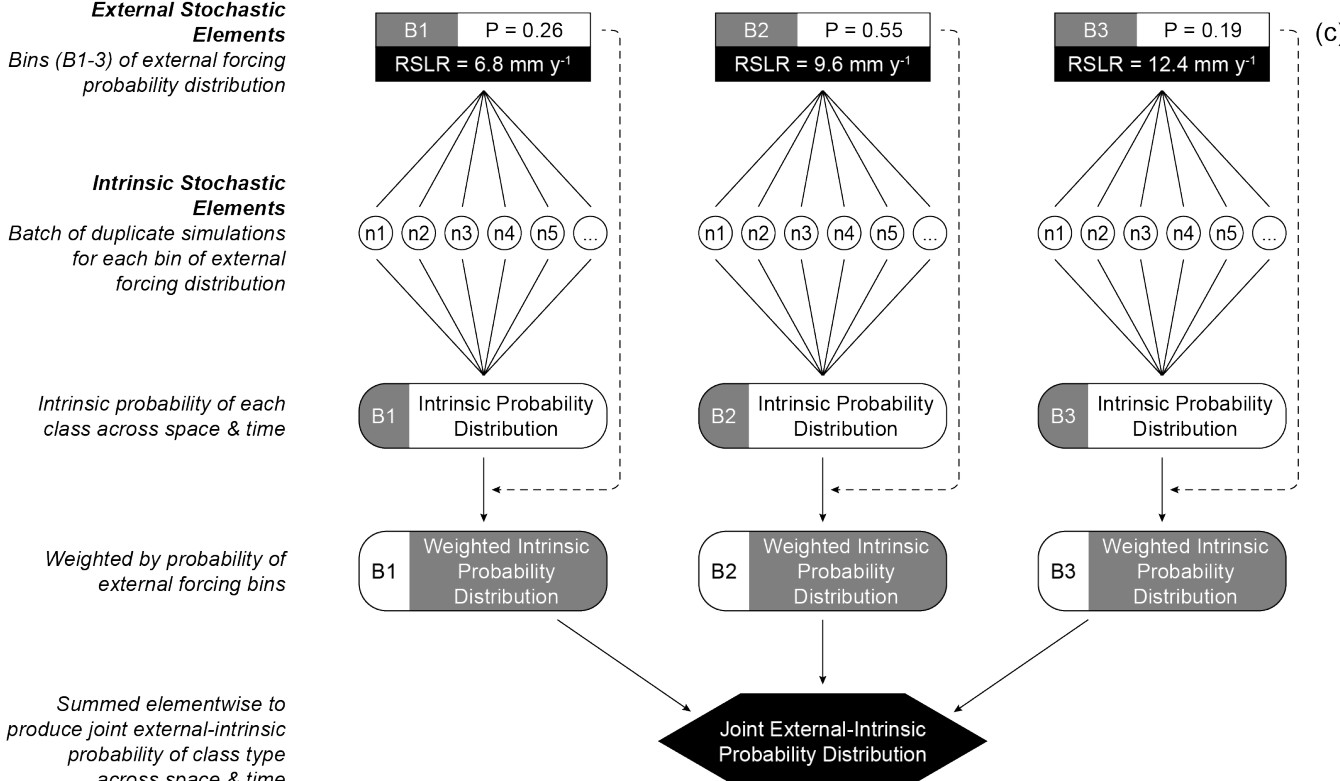

**Figure 3: Probabilistic model framework. (a) Discrete probability distribution for the rate of RSLR between 2020 and 2050 for North Core Banks, NC, USA, under IPCC SSP5-8.5. (b) Classification scheme for net elevation change used in probabilistic framework examples. (c). Annotated flowchart of the probabilistic framework in MEEB.**





For each bin of the external forcing probability distribution, the probabilistic framework in MEEB runs a batch of $n_P$ duplicate simulations and determines the class (in this case, the range of elevation change) for each cell in the domain at every

timestep for all $n_P$ simulations (Fig. 3c). These data are then used to form relative frequency distributions of class type across space and time for the intrinsic stochastic elements. Next, these distributions are weighted (multiplied) by the probability of their external forcing bin. The weighted probability distributions are then summed elementwise to produce a joint external-intrinsic probability distribution of class type for each cell of the model domain across all timesteps (Fig. 3c).

In this work, we use the likelihood of elevation change (relative to the simulation start) as an example of an outcome

that can be explored with MEEB. Elevation change is categorized into classes of major deposition, minor deposition, negligible change, minor erosion, and major erosion (Fig. 3b). Other outcomes and categorizations can be developed to suit the goals of the modeling investigation, such as the presence of vegetation or flooding (simple), or ecosystem state (more complex).

## 3 Data integration and calibration

Data integration and parameter calibration allows the user to explicitly explore the evolution of a particular setting

through time within MEEB. To ensure MEEB best represents a real-world system of interest, thorough data integration and calibration specific to each study location is critical. MEEB integrates empirical data to set initial model conditions and determine the characteristics of the forcing environment. Furthermore, given the myriad process domains, many of which employ poorly constrained parameters, comparison of simulated output to observations is used to both evaluate model performance and, importantly, calibrate process parameter values; similar calibration procedures are typical of microscale

models. Scripts for calibrating model parameters and comparing simulated results with observations are included with the model. Data integration and calibration is, of course, dependent on the availability of data, which varies significantly depending upon location and timeframe. Therefore, sources or forms of data different than described herein can be used as necessary, so long as they are processed to satisfy the same requirements detailed below.

## 3.1 Case study location

MEEB can theoretically be applied to any sandy barrier system that satisfies the minimal data requirements as described in the following subsections. As a case study, we parameterized and calibrated MEEB for North Core Banks, NC, USA using data from 2014-2018 (National Geodetic Survey, 2024; NCFMP, 2018; USACE NCMP, 2024; USDA Farm Service Agency, 2019; Ritchie et al., 2021; Sturdivant et al., 2019). North Core Banks is a sandy, wave-dominated barrier island 36 km in length, expressing a broad spectrum of ecogeomorphic states from tall, well-developed foredune fields (~5 m

above MHW) to frequently inundated overwash flats (Hovenga et al., 2021). Part of the Cape Lookout National Seashore, the barrier is minimally developed and managed to preserve endemic coastal ecosystems and natural ecogeomorphic processes. The regional rate of RSLR from 1994-2024 was approximately 6.3 mm yr$^{-1}$ (NOAA Tides & Currents, 2024) and dune vegetation is dominated by *Ammophila breviligulata,* a burial-tolerant dune-building species (Jay et al., 2022).





### 3.2 Initial conditions

MEEB requires a digital elevation model (DEM) as initial elevation input, paired with contemporaneous spatial rasters of initial vegetation density and species type (Figs. 1a, 1b). Sufficiently concurrent datasets (< 0.5 y apart) ensure that initial vegetation conditions are representative of initial elevation conditions, and vice versa. All elevation and vegetation input raster datasets were resampled to the adopted model grid resolution (here, 1 m) if necessary, clipped to a specified bounding box that defines the spatial domain, and rotated such that the domain axes are parallel with the approximate trend of the barrier

shoreline. In order to calibrate model parameter values, a minimum of two sets of observed elevation and vegetation cover datasets are needed to serve as the pre-hindcast initial conditions and the post-hindcast observations for comparison with simulation results. Our methods for satisfying these general data requirements in this case study are described in the following subsections.

### 3.2.1 Elevation

Initial topobathymetric DEMs were developed from three high-resolution lidar datasets for 2014 (NCFMP, 2018), 2017 (USACE NCMP, 2024), and 2018 (Ritchie et al., 2021). In the absence of bathymetry, and to conform with the model assumption of a linear shoreface geometry, we processed the DEMs by adding the shoreface slope by linearly increasing depth in the cross-shore dimension for all cells seaward of the ocean MHW shoreline according to $S_{sf,eq}$. A small back-barrier slope (0.05) was also added by setting the first 30 m landward of the back-barrier MHW shoreline (in the cross-shore dimension) to

increase linearly in depth from MHW to $D_{bb}$. The back-barrier cells beyond the back-barrier slope are set uniformly to $D_{bb}$.

### 3.2.2 Vegetation

We derived rasters of initial species type from supervised landcover classification and vegetation density datasets for 2014 (Sturdivant et al., 2019) and 2018 (S. Zeigler & A. Evans, U.S. Geological Survey, unpublished data, 2024) generated from orthoimagery captured within the same months as their corresponding elevation datasets. The original landcover datasets

were reclassified into three landcover classes: herbaceous vegetation, woody vegetation, and no vegetation. Using the orthoimagery from January to April 2014 (National Geodetic Survey, 2024) and October 2018 (USDA Farm Service Agency, 2019), rough approximations of initial vegetation density for the vegetated classes were derived via the Normalized Difference Vegetation Index, with thresholds set qualitatively for four classes corresponding to $\rho$ values of 0.4, 0.6, 0.8, and 1.0 with random noise perturbations of ±0.1. In qualitatively setting NDVI thresholds, we were able to roughly control for the effect of

image seasonality on the NDVI values.

### 3.3 High-water event climatology

MEEB uses a time series of modeled HWE total water level, duration, and timing for simulation hindcasts. For simulation forecasts, the observed HWE time series serves as the basis for a stochastic HWE environment, which consists of





a list of synthetic storms and a probability distribution of storm occurrence based on the time of year, as described in the
following.

### 3.3.1 Hindcast HWE time series

A modeled HWE time series (Fig. 4a) was created using hindcast hourly wave and water level conditions offshore of North Core Banks from 1979 to 2022 (Aretxabaleta et al., 2023), which include significant wave height ($H_s$), wave period ($T_p$), water level, and direction. The TWL (i.e., the representative highest elevation of the landward margin of runup) was
calculated for each hour as the sum of the maximum 2% exceedance of wave runup, following Stockdon et al. (2006), and the contemporaneous water-level elevation, which includes tides, storm surge, and other low-frequency fluctuations. As in Wahl et al. (2016) and Reeves et al. (2021), we extracted HWEs from the wave and water-level time series by conditioning upon $H_s$: HWEs were identified as periods of 8 or more consecutive hours with $H_s > 2.05$ m, which is the minimum monthly averaged wave height for periods in which water levels exceeded the 25[th] percentile of dune toe elevations (1.78 m NAVD88)
at North Core Banks measured from 2005 to 2018 (Doran et al., 2017). If two (or more) periods of 8 or more consecutive hours with $H_s > 2.05$ m occurred less than 24 hours apart, the two periods were considered part of the same large-scale weather system and therefore lumped together as single HWE; new HWEs were identified after $H_s$ remained below the 2.05 m threshold for 24 hours or longer (Li et al., 2014). Each HWE from the hindcast record is described by its maximum TWL; $H_s$ and $T_p$ concurrent with the maximum TWL; duration; and beginning and ending date/time.

### 3.3.2 Stochastic HWE environment

The timing and characteristics (TWL and duration) of HWEs are determined stochastically in model forecasts. For each Marine iteration in the model ($\Delta t_m = 0.04$ y), a HWE may occur based upon the historical probability of HWE occurrence for each iteration in the observed HWE record, which was calculated as the number of years in which one or more HWEs occurred during each Marine timestep divided by the total length (in years) of the observational record (Fig. 4b). If a HWE is
determined to occur, the TWL and duration is chosen from a list of 10,000 synthetic HWEs generated using the copula-based, multivariate sea-storm model from Wahl et al. (2016), which identifies interdependencies among relevant sea-storm parameters (water level, $H_s$, $T_p$, and duration) using the observed HWE record as required input. MEEB assumes that the probability of HWE occurrence for each iteration, as well as the characteristics of the synthetic HWEs (i.e., the average TWL and duration of the 10,000 synthetic HWEs), remain constant over the course of a simulation.

## 3.4 Wave climatology

MEEB requires estimates for long-term (multidecadal) averaged wave climate characteristics to drive alongshore shoreline diffusion, as described in Sect. 2.4.2. We used the same hindcast hourly wave and water-level conditions offshore of North Core Banks from 1979 to 2022 (Aretxabaleta et al., 2023) to derive the mean offshore significant wave height, mean



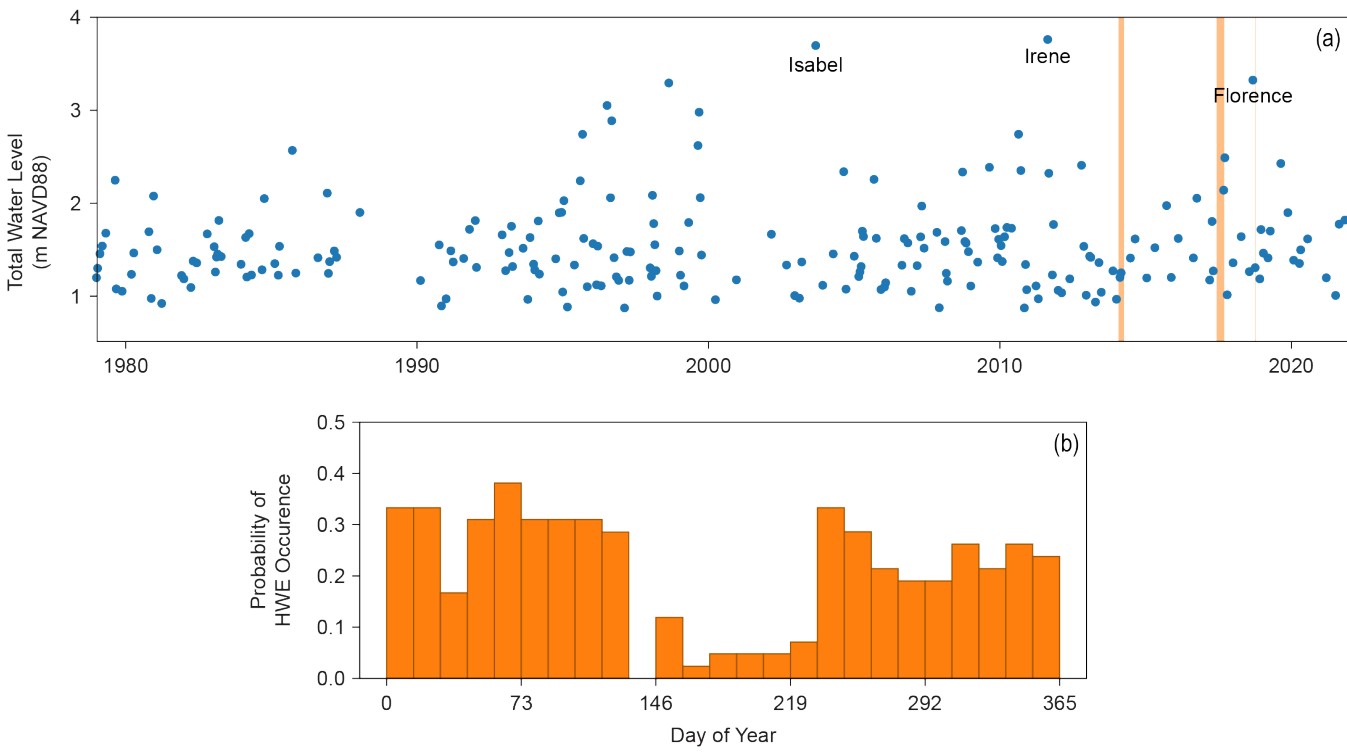

**Figure 4: Hindcast high water event (HWE) climatology for model input. (a) Total water level timeseries of HWEs (blue dots) off North Core Banks, NC, USA from 1979 to 2022; vertical orange bars indicate the dates of capture for the lidar datasets used in this study. (b) Historic (1979-2022) probability of HWE occurrence by time of year; bins are sized according to the Marine component timestep ($\Delta t_m$ = 0.04 y).**

wave period, and the wave climate asymmetry and proportion of high-angle waves (relative to the barrier shoreline trend) for the time period, which allows for computation of multidecadal shoreline change.

### 3.5 Probabilistic distribution of future RSLR

We developed a probabilistic distribution of RSLR for the year 2050 (Fig. 3a) from the IPCC AR6 SSP5-8.5 sea-level projections (Fox-Kemper et al., 2021; data available from Garner et al., 2021) for the grid tile encompassing North Core Banks at 34º N 77º W. Following others (Wainwright et al., 2014; Bamunawala et al., 2021), a triangular probability distribution of RSLR was created using the 5, 50, and 95 quantiles of the projected RSLR for the year 2050 under SSP5-8.5. This triangular distribution was then transformed into a discrete probability distribution with three bins.





SSP5-8.5 represents a "very high" greenhouse gas emissions pathway, and the accompanying sea-level rise projection follows a trajectory most similar to the NOAA "Intermediate" scenario from Sweet et al. (2022). Users can run the same probabilistic model framework with different RSLR probability distributions representative of other SSPs, and the results of multiple pathways can thereby be compared. For example, users can select and make projections for two pathways that correspond to "most likely" and "worst case" scenarios. For the year 2050, however, there is relatively little difference in projected sea levels between the highest and lowest IPCC SSPs, with considerable overlap of ranges. The relative convergence of RSLR to greenhouse gas emissions over the next three decades suggests a single SSP for a "very high" emissions pathway (i.e., SSP5-8.5) is sufficient for probabilistic projections up to the year 2050, in alignment with findings from the 2022 NOAA Interagency Technical Report for sea-level rise (Sweet et al., 2022).

**3.6 Skill assessment and parameter optimization**

We define model performance primarily through the direct cell-by-cell comparison of simulated and observed elevation with the Brier Skill Score (BSS):

$$BSS = \frac{\frac{1}{J}\sum_{j=1}^{J}(sim_j - obs_j)^2}{\frac{1}{J}\sum_{j=1}^{J}(b_j - obs_j)^2} \tag{24}$$

where $J$ is the total number of cells in the skill determination, $sim_j$ is the simulated final elevation at cell $j$, $obs_j$ is the observed final elevation at cell $j$, and $b_j$ is the baseline prediction equal to the initial elevation at cell $j$ (i.e., assumes no change). When specifically assessing the aeolian performance of the model, we also determined the BSS of the change in foredune crest elevation. While our skill assessment used a conventional cell-by-cell, point-based approach, future model calibration may benefit from alternative metrics that capture more qualitative behaviors or states (French et al., 2016; Murray, 2003).

We used particle swarm optimization (PSO) to calibrate 15 free parameters to maximize model skill. PSO is a metaheuristic computational method that searches iteratively for the global optimum within a given parameter space. PSO seeds the parameter space with particles (i.e., specific sets of parameter values) that make up a swarm (i.e., a population of candidate solutions). The movement of each particle is guided both by its own local best-known position within the parameter space as well as the best-known position of the entire swarm. Over many iterations, the swarm tends to descend upon the global optimum set of parameter values that maximizes the model skill, while tending to avoid local optima. A minimum and maximum value for each parameter defines the calibration space; see Table 1 for the range of values used for each calibrated free parameter.

We optimized aeolian and marine parameters separately to reduce the number of free parameters calibrated at once and improve potential calibration. First, marine parameters alone were optimized over a single HWE event: Hurricane Florence, which made landfall as a Category 1 hurricane in September 2018 south of Wrightsville Beach, NC, USA, was used





for marine calibration because of extensive overwash on North Core Banks and the availability of lidar captured 3 weeks after the event (Ritchie et al., 2021). We used a skill score for change in elevation that was designed to be representative of the

barrier as a whole, calculated as the average BSS of elevation change across five ecogeomorphologically diverse, 300-m-long training sites characterized by a small and large overwash fan, a small and large overwash flat, and a tall continuous dune ridge (Figs. 5, 6). To best ensure that the observed change is representative of HWE impacts, determination of skill was limited to subaerial cells that fall seaward of the dune crest or within either the simulated or observed overwash extent. Observed overwash extent was digitized by hand with pre- (Jun to Sep 2017; USACE NCMP, 2024) and post-Florence (Oct 2018;

Ritchie et al., 2021) imagery and lidar as reference.

Next, aeolian parameters were optimized by running the model over a period of relatively limited HWE activity (16 April 2014 to 16 September 2017), thereby controlling for the effects of marine processes on the aeolian calibration. Results from the preceding marine parameter calibration were used to set the marine parameter values for the aeolian calibration. The RSLR rate was set to 6 mm yr$^{-1}$ for these hindcasts, representative of historical conditions. We used a multi-objective skill

score calculated as the average of the BSS for cell-by-cell elevation change and the BSS for dune crest height change. Including dune height as part of the multi-objective skill score helps prioritize accurate representation of vertical foredune growth over other characteristics, particularly as foredune height has a large impact on barrier response to HWEs. To focus calibration on foredune growth and to exclude interior areas with error in the lidar DEMs associated with reflectance off the vegetation surface, determination of skill was also limited to cells within a specified foredune field (Fig. 6). This multi-objective score

was averaged again across three diverse, 200-m-long training sites to produce a single score representative for North Core Banks as a whole. To calibrate the four wind-direction probabilities ($P_{WD}$) that collectively sum to 1, we used three parameters that can be tuned independently from each other: 1) the wind axis ratio, which is the ratio of wind directed cross-shore relative to alongshore; 2) the onshore wind ratio, which is the ratio of cross-shore wind directed onshore relative to offshore; and 3) the down-shore wind ratio, which is the ratio of alongshore wind directed down the coastline relative to up. The four values of

$P_{WD}$ were calculated from the three resulting calibrated wind parameters. While historical wind direction data from North Core Banks is available for input into MEEB, we found calibrating the wind direction probabilities directly produced significantly better agreement with observations. This may be because the dominant historical wind directions at North Core Banks run oblique to the shoreline and therefore our model grid orientation, coupled with the fact that aeolian transport in the model is constricted to directions directly parallel with model gridlines (i.e., not diagonal), suggesting limitations to the relatively simple

aeolian algorithm employed in MEEB (see also Nield & Baas, 2008a). Future research into incorporating wind direction observations, particularly with oblique wind directions, could prove fruitful.

Due to challenges in generating effective skill scores for hindcasts of vegetation cover, we did not calibrate vegetation free parameters; instead, vegetation parameter values roughly follow those of Nield and Baas (2008a) and Keijsers et al. (2016). Shoreline parameters, with perhaps the exception of the alongshore sediment transport constant $k_d$, can be directly estimated

and therefore do not necessitate optimization.





**Figure 5: Hindcast simulations testing MEEB performance with calibrated marine parameters. (a-d) Observed and simulated post-Florence elevation and pre- to post-Florence elevation change at 4 test locations across North Core Banks, NC, USA. Locations of each marine testing site, as well as each marine training site used in calibration, are indicated in the top map. Skill scores for each hindcast (a-d) are given in Table 3.**





Figure 6: Hindcast simulations (for the period of April 2014 to September 2017) testing MEEB performance with calibrated aeolian parameters. (a-d) Observed and simulated post-simulation elevation and pre- to post-simulation elevation change at 4 test locations across North Core Banks, NC, USA. Locations of each aeolian testing site, as well as each aeolian training site used in calibration, are indicated in the top map. Skill scores for each hindcast (a-d) are given in Table 3. Skill determination is confined to the area



**between the dashed lines at each location to focus calibration on foredune growth and exclude error in the lidar DEMs associated**
**with reflectance off the vegetation surface. Areas of apparent significant accretion landward of the foredune field in panels a-d are artefacts of the post-Florence lidar capturing the shrub canopy elevations versus the true surface elevation in the pre-Florence lidar.**


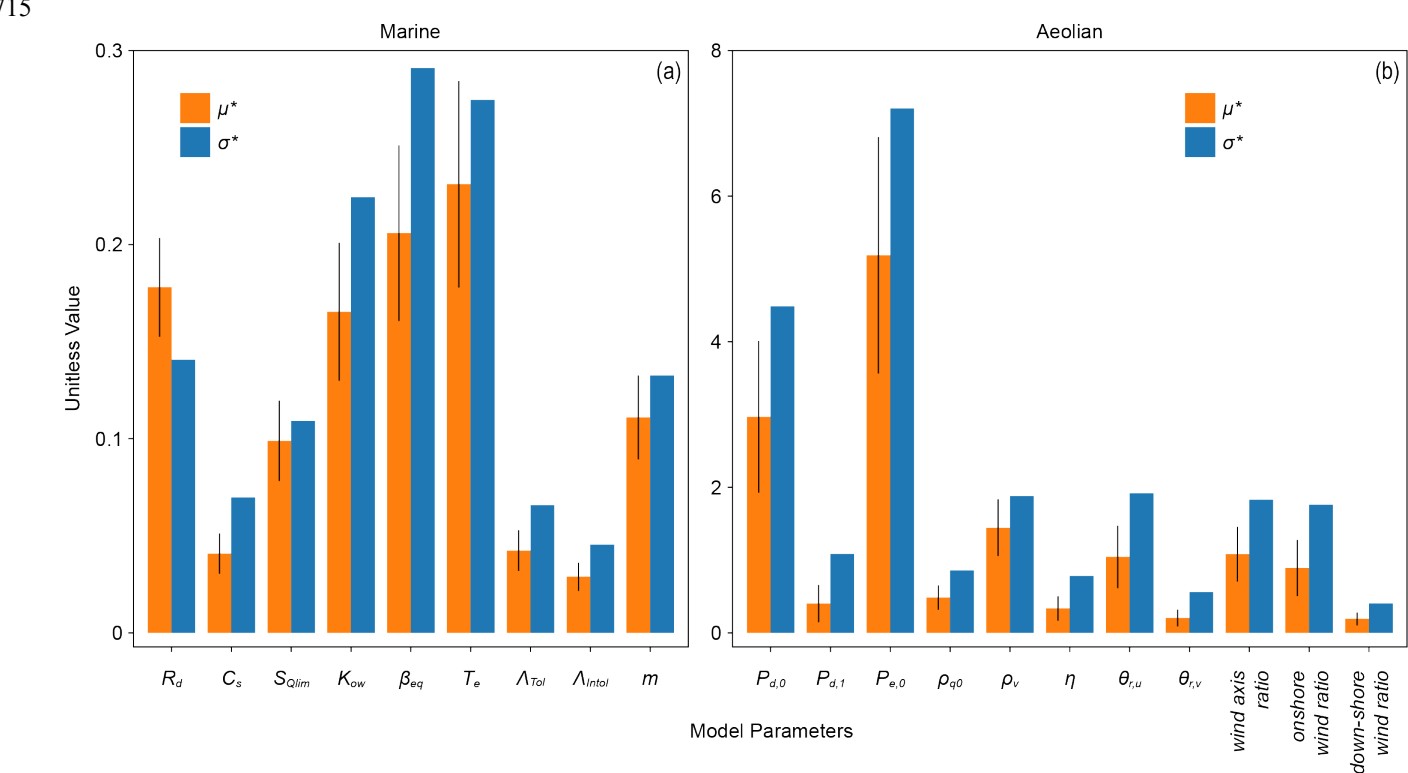

**Figure 7: Sensitivity analyses of model parameters for the (a) Marine and (b) Aeolian model components. Marine and aeolian parameters are analyzed separately following the same design and parameter ranges as the parameter calibration simulations.**
**Unitless values for the overall importance of a parameter ($\mu^*$) and its degree of interaction with other parameters ($\sigma^*$) are given in orange and blue, respectively. Black lines represent the 95% confidence interval.**





### 3.7 Sensitivity analyses

We investigated the global sensitivity of model output to input parameter variations for the purpose of ranking
variables according to their relative contribution to output variability and screening variables that have minimal effect on
output variability. We used the Method of Morris test (Morris, 1991), also known as the Elementary Effects Test, which is
well-suited for models like MEEB with numerous input variables and/or relatively long run times (Pianosi et al., 2016). The
Method of Morris determines both the overall importance of a parameter ($\mu^*$; the mean of the absolute value of the elementary
effects) and its degree of interaction with other input parameters ($\sigma^*$; the standard deviation of the absolute value of the
elementary effects). As in the calibration workflow, we performed sensitivity analyses on the Aeolian and Marine components
of the model separately, and used the same experimental setup, skill scores, and input parameter ranges. Because the Method
of Morris depends on the range of input parameter values and because several parameters in MEEB are poorly constrained and
abstract, conclusions drawn from our sensitivity analyses are relevant only with regards to the estimated ranges of parameter
values used herein (Table 1). Our sampling strategy used six grid levels (which controls the sampling grid and variation sizes;
Morris, 1991; Pianosi et al., 2016), with 75 aeolian and 115 marine trajectories resulting in 900 and 1150 combinations of
parameter values evaluated for the Aeolian and Marine components of the model, respectively.

## 4 Example simulations and results

### 4.1 Parameter sensitivity analyses

Our parameter sensitivity analyses suggest that the Marine component of MEEB is most sensitive to the erosive
timescale calibration coefficient ($T_e$) and equilibrium beach slope ($\beta_{eq}$), followed by the overwash infiltration and drag
parameter ($R_d$) and sediment transport coefficients ($K_{ow}, m$; Fig. 7a). These parameters also display the highest degree of
interaction with each other (Fig. 7a). The Marine component on MEEB is insensitive to the vegetation flow-reduction
coefficient ($\Lambda$), which controls the degree to which vegetation impacts the overwash flow, for both types of vegetation; we
therefore did not include these parameters in our final calibration. Perhaps unsurprisingly, marine evolution in the model is
particularly sensitive to poorly constrained coefficients ($T_e, K_{ow}, m$) that have limited relevance to real-world measurements
or observations. That overwash in the model is particularly sensitive to infiltration and drag of the flow across the barrier
interior ($R_d$) suggests continued study and measurement of these factors across the barrier landscape could be especially
beneficial.

The Aeolian component of MEEB is most sensitive, by far, to the probabilities of deposition ($P_{d,0}$) and erosion ($P_{e,0}$)
in the absence of vegetation (Fig. 7b). These parameters also display the highest degree of interaction with other parameters.
The model is insensitive to both the unvegetated and vegetated angles of repose ($\theta_{r,u}, \theta_{r,v}$), and we therefore exclude them
from the model optimization. Overall, aeolian evolution in the model is most sensitive to parameters controlling the volumetric
sediment flux, as opposed to the exact specifications of the way in which vegetation density and type alter this flux.



### 4.2 Hindcasts: comparisons to observations

To assess model skill, we ran hindcast simulations using the calibrated parameter values and the same simulation set-up as in calibration except at testing sites that differ from the training sites (Figs. 5, 6). Simulation results were compared to observed elevation change and the resulting model skill scores are given in Table 3.

Hindcast simulations of Hurricane Florence using calibrated marine parameters produce good to excellent agreement with observations (following the BSS categorization from Sutherland et al., 2004) for pre- to post-storm elevation change at
four test sites across North Core Banks (Fig. 5; Table 3). Good to excellent agreement is also found when considering elevation change landward of the foredune crest in isolation, as well as fair to excellent agreement seaward of the foredune crest (Table 3). MEEB does particularly well in capturing overwash deposition patterns and washover thicknesses. Naturally, the model is more skillful at some test sites across the barrier than others. In particular, the model tends to overestimate dune scarping in certain areas, particularly where overwash flows through gaps in the foredune line (Figs. 5a, 5c). MEEB also tends to
underestimate lateral spreading of overwash flow in areas with confined or channelized topography (Fig. 5b), a direct consequence of the simplified flow-routing algorithm that directs flow only to the three landward neighboring cells.

Testing the skill of calibrated aeolian parameters, hindcast simulations from April 2014 to September 2017 (a period of minimal storm activity) at four test sites across North Core Banks result in good to poor agreement with observed topographic change (Fig. 6; Table 3). As in calibration, the skill assessment was applied only to cells within the predetermined
dune field (Fig. 6). While beach change is predicted poorly in some of the hindcasts (Figs. 6b, 6c), it is beyond the intent of the model to predict fluctuations in beach state on a monthly or seasonal timescale. Importantly, the model appears to do qualitatively well in capturing the location and/or pattern of dune growth. This includes the one hindcast with a poor (0.04) BSS (Fig. 6b), suggesting that our simple multi-objective BSS could be improved to better identify and prioritize important qualitatively correct behaviors (French et al., 2016) and better align with subjective judgements of similarity (Bosboom and
Reniers, 2014). In our tests, the model tends to underestimate the vertical extent of dune growth. Additionally, aeolian reworking of the barrier interior (landward of the foredune crest) is overestimated in some areas (Figs. 6a, 6d), suggesting that the model is sensitive to the initial vegetation conditions.

### 4.3 Forecasts: probability of future change

MEEB was developed to provide useful projections at spatiotemporal scales relevant to coastal management. Here,
the model is exercised in a predictive application using the probabilistic framework described in Sect. 2.6 and the RSLR probability distribution described in Sect. 3.5. We run example probabilistic projections of elevation change for the year 2050 at an initially overwash-prone site and an initially overwash-resistant site on North Core Banks. Fig. 8 plots the most likely range of net elevation change across the model domain from 2018 to 2050 for these projections. At the initially overwash-prone site, model projections suggest that major deposition is most likely at the proximal parts of the overwash fans with minor
deposition most likely on the more distal portions (Fig. 8a). The model also predicts the high likelihood of major accretion



**Table 3: Model skill scores for elevation change from MEEB hindcast simulations. BSS = cell-by-cell Brier Skill Score; BSS_Seaward = BSS of all cells seaward of the post-storm foredune crest; BSS_Landward = BSS of all cells landward of the post-storm foredune crest; BSS_DuneElev = BSS of change in elevation of all foredune crest cells; Multi_Obj_BSS = average of BSS and BSS_DuneElev. Brier Skill Score qualitative classification follows Sutherland et al. (2004).**

| Marine | | | | |
|---|---|---|---|---|
| **Location** | **Classification** | **BSS** | **BSS_Seaward** | **BSS_Landward** |
| Fig. 5a | Excellent | 0.65 | 0.69 | 0.46 |
| Fig. 5b | Excellent | 0.64 | 0.64 | 0.40 |
| Fig. 5c | Good | 0.22 | 0.19 | 0.20 |
| Fig. 5d | Excellent | 0.79 | 0.79 | 0.56 |
| Aeolian | | | | |
| **Location** | **Classification** | **BSS** | **BSS_DuneElev** | **Multi_Obj_BSS** |
| Fig. 6a | Good | 0.24 | 0.04 | 0.14 |
| Fig. 6b | Poor | 0.03 | 0.03 | 0.03 |
| Fig. 6c | Good | 0.23 | 0.22 | 0.23 |
| Fig. 6d | Good | 0.23 | -0.03 | 0.10 |


around the seaward slope and toe of the present foredunes, reflecting the steeping of the beach profile and net seaward

expansion of the foredune system. Aeolian transport from sparsely vegetated overwash fans resulting in minor deposition along

the landward vegetated fringes (cf. Rodriguez et al., 2013) is also predicted to be likely. This probabilistic projection suggests

that the areas encompassing the washover fans are likely to remain vulnerable to persistent overwash through the year 2050,

while foredune areas are likely to not only be stable but expand.

At the initially overwash-resistant site, probabilistic projections suggest that major lateral dune erosion via scarping

is likely to occur but that the foredune ridge will most likely persist (Fig. 8b). Aeolian deposition near the initial foredune crest

is likely to offset some of the height and volume lost from dune scarping. Although the model predicts negligible elevation

change is most likely landward of the foredune crest, uncertainty in this prediction – particularly near the left (Southwest) edge

of the model domain – reflects the small but very real possibility of minor to major overwash deposition in the event of

foredune overtopping and/or loss (Fig. 8b). Overall, this projection indicates that vulnerability to HWE-driven change is low

through 2050 landward of the initial 2018 foredune crest, though the high likelihood of major dune width loss in this period

suggests that the vulnerability of the barrier interior to overwash may increase rapidly in the subsequent decades.



**Figure 8: Example probabilistic projections of elevation change for the year 2050, at two sites on North Core Banks, NC, USA: (a-c) an initially overwash-prone site and (d-f) an initially overwash-resistant site. Locations of both sites are indicated in the top map.**




**(a, d) Initial 2018 topography for the overwash-prone and overwash-resistant sites, respectively. (b, e) Most likely net change in elevation between 2018 and 2050 classified into five elevation ranges. (c, f) Certainty in most likely elevation change predictions, taken as the proportion of all model runs resulting in the most likely class of elevation change.**

## 5 Discussion and conclusions

MEEB was developed to simulate spatially explicit, ecogeomorphic, probabilistic evolution of coastal barrier systems over spatiotemporal scales of greatest interest to coastal managers and decision-makers. The goal of the model is to reconcile management needs for projections that are both quantitative/place-specific and multi-decadal/multi-kilometer (i.e., mesoscale) while accounting for ecogeomorphic feedbacks and uncertainties in the forces driving future change. In our approach for modeling meso-scale barrier ecogeomorphic evolution, we a) integrated model parameterizations of varying mechanistic complexity, representing certain processes or interactions with relatively higher complexity (only as far as needed to produce mesoscale behaviors anticipated to be important) within an otherwise synthesist framework, and b) thoroughly tested and calibrated these parameterizations with observational data. Because many of these relatively simple and heuristic algorithms involve calibration coefficients and poorly constrained independent variables, we have made a substantial effort to determine which of these parameters are most influential to model outcomes and assign values that provide the best model agreement with observations, as evaluated by our multi-objective skill scores for hindcasts on North Core Banks. While the model has yet to be applied to other sites, North Core Banks is an ideal case study location given its relative lack of human influence, multiple ecosystem types and barrier states, history of ecogeomorphic change, and availability of high-quality data, and therefore may be representative of other unmanaged barrier environments. Given the demonstrated skill of our hindcasts of North Core Banks, as well as our holistic and simplistic model representations of ecogeomorphic processes and interactions, we expect MEEB to be widely adaptable to most unmanaged barrier systems along the U.S. East and Gulf coasts. However, the model may be less applicable to barrier environments strongly influenced by tidal inlet dynamics, complex shoreline change, or human management, though in the future a human management module could be added to MEEB (e.g., Anarde et al., 2024a,b) to explore the ecogeomorphic effects of coupled human-natural interactions.

As with any numerical model, appropriate interpretation of the results depends on the scale and complexity of the model parameterizations. As a mesoscale model, MEEB is not suitable for investigations with the goal of predicting subtle changes in elevation or vegetation cover; nor is it an appropriate tool for explaining the large-scale behaviors of landscape configuration, such as the drowning of a barrier chain. Instead, MEEB is designed to answer questions about barrier ecogeomorphic change of moderate complexity by offering semi-qualitative predictions and semi-quantitative explanations. For example, MEEB can be used to investigate the effects of climate-induced shifts in ecological composition by adjusting vegetation parameters over time to represent transitions in vegetation assemblages. Such transitions could affect the likelihood of coastal hazard impacts by altering aeolian growth rates and/or patterns and therefore landscape vulnerability to HWEs. As another example application, MEEB could be used to predict where and when overwash along a barrier is most likely to occur





by generating probabilistic projections of overwash inundation through time and identify ecogeomorphic indicators of
overwash vulnerability.

To simulate spatially explicit ecological and geomorphological processes across an entire subaerial barrier landscape over several decades, MEEB employs many synthesized parameterizations and simplifying assumptions. Inevitably, MEEB bears several important limitations arising from this modeling approach. For example, MEEB assumes the barrier system is composed entirely of unconsolidated sand, whereas grain sizes and characteristics can vary over meters to kilometers in ways
that affect dune growth (e.g., Hovenga et al., 2023) and barrier transgression (e.g., Brenner et al., 2015). Tidal inlet and breaching processes, including outwash events (Sherwood et al., 2023), are neglected in MEEB, despite the significant ecomorphodynamic changes associated with these processes and their importance to long-term transgressive sediment flux (e.g., Nienhuis and Lorenzo-Trueba, 2019b; Leatherman, 1979; Passeri et al., 2020; Sherwood et al., 2023). Therefore, we exclude portions of a barrier within several hundred meters of active tidal inlets and breaches from the model domain, rendering
the model less relevant to tide-dominated barrier systems where barriers are frequently segmented by inlet channels. Further, MEEB models the long-term, wave-climate-averaged shoreline diffusivity by assuming the diffusive wave climate does not change over the course of a simulation, which may average over variability in shoreline diffusion that could potentially feedback with coupled ecogeomorphic processes (e.g., dune erosion). MEEB also presently assumes that no shoreline changes resulting from alongshore transport gradients occur at the domain boundaries (i.e., shoreline change at these cells only occurs
from cross-shore processes). As such, if the MEEB model domain were part of a curved coastline eroding or accreting due to persistent alongshore transport gradients (Ashton & Murray, 2006), local shoreline change rates would be underpredicted. Lastly, the Vegetation component presently lacks finer temporal resolution needed to capture seasonal variations in vegetation density and growth, which could play an important role in how a barrier responds to and recovers from HWEs. Additionally, the simplified Vegetation component lacks many of the environmental filters (e.g., temperature, groundwater depth, elevation,
salinity) that could affect the zonation and density of vegetation, with implications for aeolian and overwash sediment transport and thereby projections of future change. To improve the utility of MEEB for addressing the ecogeomorphic effects of climate-induced shifts in ecology, the model would require the addition of seasonality and species zonation dynamics.

Despite these limitations, comparisons of MEEB hindcast simulations to observations demonstrate that our relatively simple set of parameterizations for coupled aeolian, marine, vegetation, and shoreline processes can skillfully capture
important mesoscale dynamics of barrier systems. This is achieved in large part through thorough calibration of the most sensitive free parameters, with integration of topographic, ecologic, and storm climatology data to set initial conditions and assess model performance. A single set of calibrated parameter values performs well at ecogeomorphologically diverse test sites spread across the nearly 30 km domain and in hindcast simulations spanning multiple continuous years, leading to confidence in model projections of future mesoscale change.



**Code availability**

MEEB source code and documentation are available for download from Reeves (2024) under a CC0 1.0 Universal license.

**Data availability**

Hindcast hourly wave and water level conditions offshore of North Core Banks from 1979 to 2022, used to develop
the observed HWE timeseries and stochastic HWE environment, are available from the Renaissance Computing Institute (Blanton et al., 2024) and stored within the MEEB model directory at /MEEB/Data/NorthCoreBanks.

**Author contributions**

IRBR developed the model code, prepared input data, ran the simulations, analyzed the results, and wrote the original draft of the manuscript. All authors contributed to the project conceptualization, methodology, and editing of the manuscript.

**Competing interests**

The authors declare that they have no conflict of interest.

**Disclaimer**

Any use of trade, firm, or product names is for descriptive purposes only and does not imply endorsement by the U.S. Government.

**Acknowledgements**

This work was supported by a Woods Hole Oceanographic Institution and U.S. Geological Survey Postdoctoral Scholar Fellowship, and the U.S. Geological Survey Coastal and Marine Hazards and Resources Program as part of the Remote Sensing Coastal Change and Future Landscape Adaptation and Coastal Change projects. IRBR was supported in part by the U.S. Geological Survey's John Wesley Powell Center for Analysis and Synthesis as part of a working group entitled "Beyond
waves and shifting sand: considering ecosystem processes in forecasts of coastal ecosystem change;" this project was co-sponsored by the National Science Foundation under grant 23DEB10330. Wave and water-level model hindcasts used in this work were provided by the Renaissance Computing Institute.





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
