# Peer review of "Projecting management-relevant change of undeveloped coastal barriers with the Mesoscale Explicit Ecogeomorphic Barrier model (MEEB) v1.0"

_Geoscientific Model Development, 2024_

## Referee Comment (RC2)

[revised manuscript text omitted]

$$S_{sf} = \frac{D_{sf}}{x_s - x_t}$$  (11)

where $D_{sf}$ is the shoreface depth. The shoreface slope is allowed to deviate from its equilibrium slope ($S_{sf,eq}$) in response to perturbations. When the shoreface steepens past its equilibrium configuration (e.g., as a result of RSLR; Bruun, 1962), shoreface fluxes are directed offshore; if the shoreface shallows past its equilibrium (which can occur when overwash and aeolian processes remove sediment from the upper shoreface), shoreface fluxes are directed onshore. Shoreface flux ($q_{sf}$) therefore depends on the deviations of the shoreface slope from its equilibrium:

$$q_{sf} = k_{sf}(S_{sf} - S_{sf,eq})$$  (12)

*(handwritten: + offshore. Is this the sign convention? I could have thought + when land being gained.)*

[revised manuscript text omitted]

---

## Author Response (AR1)

**Response to Reviewer Comments**

for

**gmd-2024-232: Projecting management-relevant change of undeveloped coastal barriers with the Mesoscale Explicit Ecogeomorphic Barrier model (MEEB) v1.0**

We are grateful to Dr. Lazarus and Dr. Wickert for their generous feedback which has helped us significantly improve our manuscript. Below, we provide our responses (indented) to referee comments (in bold). All line numbers refer to the original submitted manuscript.

**REVIEW 1**

This contribution describes a numerical model designed to address mesoscales of physical and ecological coastal processes "over years to decades and hundreds to thousands of meters and with meter-scale spatial resolution" [L78]. The authors explain that the model is "designed to answer questions of moderate complexity regarding when, where, and how ecogeomorphic change is likely to occur, with correspondingly moderate levels of both predictive (quantitative) and explanatory (qualitative) power" [L140].

I found the manuscript clear and comprehensive in its explication — which are the qualities a potential user needs most from a model description. The demonstrations of short (three-year) and long (multi-decadal projections) time scales are an interesting exercise.

We appreciate this positive feedback.

From the results presented (Figs. 5, 6, 8), it appears that the model largely reiterates topographic controls on morphological change, which makes me wonder what kind of forcing would be required for the model to predict a real shift in morphological regime: from overwash-prone to overwash-resistant, for example, or vice versa (Fig. 8).

We appreciate this question. For the probabilistic projections (Fig. 8), the model predicts that these two locations will most likely be relatively stable over the next couple decades. Given our use of observational data to set initial conditions and forcings and carefully calibrate model parameters, we of course consider this a model prediction rather than artifact. That being said, shifts in morphological regime do appear to have occurred in a minority of the 96 batch simulations that comprise the probabilistic projections (Fig. 8); we can see this particularly in Figs. 8e and 8f, where negligible elevation change is most likely to occur landward of the dune crest (Fig. 8e), but this most likely prediction is not fully certain (Fig. 8f), suggesting there is a smaller probability of significant dune loss and overwash. The loss of dune width from 2024 to 2050 also suggests that a shift in

morphological regime (from overwash-resistant to overwash-prone) may become the most likely prediction in the decades following 2050. We have improved this explanation in the manuscript (L804):

"Overall, this projection indicates that vulnerability to HWE-driven change is low through 2050 landward of the initial 2018 foredune crest, though the high probability of major dune width loss in this period suggests that the likelihood of a shift in morphologic regime from overwash-resistant to overwash-prone may increase rapidly in the subsequent decades."

Therefore, shifts in morphologic regime at these two locations would be more likely to occur with bigger or more frequent HWEs, slower aeolian recovery, and/or more time. We now mention that changes to HWE intensity could make regime shifts more likely in our projections (L806):

"Potential increases in future HWE intensity (e.g., Knutson et al., 2020) could also enhance the likelihood of more fundamental morphological and ecological regime changes by 2050 – such fundamental changes would also be likely to occur by the end of the century."

A couple of very minor remarks. First, I think including the skill results somewhere beside or within the paired "observed" and "simulated" panels (Figs. 5, 6) would be helpful for the reader to interpret what they're already comparing visually. The skill results can also be reported in a table, as they are presently – this is a both/and suggestion.

Thank you for the suggestion. We have added the skill scores to each panel in Figs. 5 & 6 and have updated the captions accordingly.

Second, although the stated intention of the model is to forecast "when, where, and how ecogeomorphic change is likely to occur", the multi-year results (Figs. 5, 6) and probabilistic projections (Fig. 8) show elevation change. It's not immediately clear how or where vegetation manifests in those elevation maps. I raise this only to point out that the maps, as presented, appear to reflect geomorphic change; the "eco" here could perhaps be made more explicit. The authors have been careful with their caveats in the kinds of forecasts this model delivers: the spatio-temporal changes in vegetation characteristics (density, patchiness) are likely to be more qualitative here, but quantifying extents and/or trends of change across the model domain under a given set of conditions might nonetheless be informative.

We agree that ecological dynamics in the example model simulations we provided were not as explicitly highlighted as the geomorphic dynamics. This is partially because the model hindcasts from Fig. 6 only run for about 3.5 years over a period with relatively little disturbance, thus the vegetation change is relatively minimal. For the 32-yr

probabilistic forecast examples (Fig. 8), however, we have added text to Section 4.3 that more explicitly describes the ways in which vegetation tends to dynamically change and influence geomorphic evolution in the forecasts:

"At the initially overwash-prone site (Figs. 8a-c), model projections suggest that major deposition is most likely at the proximal parts of the overwash fans with minor deposition most likely on the more distal portions (Fig. 8b). Repeated overwash events will tend to prevent vegetation from recolonizing the overwash fans over the course of the simulation. Consequentially, aeolian deflation of the sparsely vegetated overwash fans, and resulting minor deposition along the landward vegetated fringes of the fans (cf. Rodriguez et al., 2013), is also predicted to be likely. The model also predicts the high likelihood of major accretion around the seaward slope and toe of the present foredunes, reflecting the steeping of the beach profile with net seaward growth of the foredune system and likely net seaward expansion of vegetation cover." (L783)

"At the initially overwash-resistant site (Figs. 8d-f), the probabilistic projection suggests that major lateral dune erosion via scarping is likely to occur but that the foredune ridge will most likely persist (Fig. 8e). Aeolian deposition near the initial foredune crest is likely to offset some of the height and volume lost from dune scarping. As a result of this persistent and resistant topography, dense vegetation will tend to cover the barrier interior and prevent aeolian reworking landward of the dune crest." (L799)

"Potential increases in future HWE intensity (e.g., Knutson et al., 2020) could also enhance the likelihood of more fundamental morphological and ecological regime changes by 2050 – such fundamental changes would also be likely to occur by the end of the century." (L806)

Finally, it strikes me that the examples provided here offer "meso" time scales but spatial scales more aligned with the models the authors characterise as "microscale". All the domains shown are for a barrier reach of 500 m; I am curious about what model outputs look like at spatially extended scales. I can understand how the spatial scale used here serves the purpose of demonstration, but a 500 m domain ultimately seems slightly misaligned with the motivation of the Introduction. Do the skill scores go down as spatial scales increase? (That is, to what extent does the user trade skill for scale?) Could skill scores somehow be normalised by spatial scale? (Is it ever reasonable to expect high-resolution predictive precision over many kms?)

We agree that the examples provided, which span only 0.5 km in length alongshore, do not fully demonstrate the spatial scales the model is specifically designed to simulate. However, as noted by the reviewer, a smaller domain enables us to more clearly and concisely demonstrate and explain the model output, which we feel is the bigger priority. Therefore, we have added this justification and an explanation that the model can handle much larger domain sizes (L782):

"While these sites span only 0.5 km in length alongshore for the purpose of providing a clear and concise demonstration of model output, MEEB can handle model domains up to tens of kilometers in alongshore length."

Skill scores do not necessarily reduce with increasing domain size, as some locations of a barrier would score higher, while others lower. Therefore, skill scores would tend to approach the mean score of the entire barrier as the domain extent is increased. We have added this explanation to the manuscript (L757):

"Our testing sites each span 0.5 km in length alongshore to demonstrate the variability of model performance in different geomorphic settings; with increasingly larger model domain extents, the skill scores would tend to approach the mean score of the entire barrier."

Perhaps in future work – because I'm sure it's outside the scope of this effort – the authors could, for a selected barrier site, push a microscale model (e.g., XBeach) up to its maximum spatio-temporal limits and push this model from its minimum spatio-temporal limits, to explicitly illustrate and examine where, when, and how the trajectories of the micro/meso models cross over in their respective utility.

This is a thoughtful idea. We have added a sentence in the Discussion introducing it as potential future work (L845):

"Future work could compare MEEB simulations with micro- (e.g., XBeach, Roelvink et al., 2009) or macro-scale (e.g. LTA14, Lorenzo-Trueba and Ashton, 2014) models to explicitly determine when, where, and how the trajectories of the models overlap in their respective utility."

I look forward to seeing this work in print in GMDD.

**REVIEW 2**

I agree with the authors that the need for such meso-scale models is essential for applied geomorphology and prediction of changes in landscapes and hazards into the coming decades. I find their manuscript to be well-written, well-illustrated, and a strong candidate for publication.

I attach a PDF containing an extensive set of handwritten comments on this paper. Some of these comments are related to writing and communication. Many link to questions about

the science, its clarity (generally good in this paper, notwithstanding), and communication of the methods.

Because of the strength of this paper alongside the rather large number of moderate changes or clarifications I have suggested, I recommend that the authors provide minor revisions.

Thank you for the positive feedback. In our revision, we have adopted all fixes to grammar, style, and typos/errors suggested in the annotated PDF. Additionally, we have responded to each handwritten comment in the PDF, including (amongst others):

• Adding additional details on computer specs needed to run the model (L153):

"The model is written in Python and can be run on PC, Macintosh, and Linux operating systems. The typical runtime for a 10-year simulation of a 1-km-long barrier segment with 1-m grid resolution is approximately 40 to 80 min, or approximately 10 to 20 min with a grid resolution of 2 m. Memory usage depends strongly on domain size, grid resolution, and the frequency and type of saved model output. An individual deterministic simulation in MEEB is run on an individual core, while our probabilistic framework runs batches of deterministic simulations in parallel across multiple cores (as many as allocated); a high-performance computing cluster is recommended for probabilistic simulations spanning >10 km of shoreline."

- Annotating Fig. 1d with relevant variables
- Moving Table 1 and 2 to the Appendix, and renaming Table 3 as "Table 1"
- Adding a short justification for the approximation of the groundwater surface (L232):

"Assuming that the groundwater surface typically resembles a subdued reflection of the topography, the groundwater surface in MEEB..."

We also note that it is unnecessary to flatten the water surface of ponds in the model because  $P_e$  and  $P_d$  will equal 0 and 1 (respectively) regardless of whether ponds are flattened or not (L234):

"Groundwater can intersect topographic depressions as surface ponds (MEEB does not flatten the water surface in ponds given that  $P_e = 0$  and  $P_d = 1$  regardless)."

• Reordering the second paragraph of Section 2.3 (L250):

"Every 25-1" y ( $\Delta t_m = 0.04$  y or ~14.6 d), MEEB determines whether a HWE occurs depending on the observed time series (for hindcasts) or a probability of occurrence dependent on the time of year (for forecasts). If no HWE is determined to occur for a Marine iteration, no marine processes take place (i.e., the landscape remains unaltered) and MEEB proceeds directly to the Shoreline component of the model. If a HWE is determined to occur for a Marine iteration, the HWE is described by a total water level (TWL)..."

• Clarifying  $q_x$  in Eq. 2 as a deposited volume and therefore porosity is implicitly included (L277):

"the deposited volume of sediment at cross-shore location x,  $q_x$ , is equal to..."

• Noting the derivation/source of the overwash flow routing constant n (L314):

"...and n is a constant equal to 0.5 (derived from the equation for motion of uniform flow; Murray and Paola, 1997)."

• Clarifying variable Qs as a volumetric sediment flux (volume per cell per iteration) in the overwash flow routing section (L322)

"The depositional volume of sediment transported each iteration (i.e., the volumetric sediment flux) from the distributing cell to landward neighbor i..."

• Emphasizing the physical basis for the exponential decay distribution of overwash sediment delivered to the subaqueous back-barrier environment (L331):

"Where overwash reaches the back-barrier shoreline, the sediment load into the subaqueous back-barrier environment is distributed in an exponentially decaying fashion, with the landward neighbor with the most discharge receiving the most sediment, which produces steeply dipping delta-like foreset deposits typically observed when overwash flows into standing bodies of waters (Schwartz, 1982; Shaw et al., 2015)"

• Adding a description of the simple temporal discretization method used to avoid instabilities in the Marine component of the model (L346):

"Our method involves simply dividing the resulting elevation change at each substep by the number of substeps within the hour."

• Adding a figure (Fig. 4c) to show how shoreline diffusivity depends on shoreline angle for the given long-term wave climate of North Core Banks (L453):

"The nonlinear dependence of shoreline diffusion on wave angle mostly affects the overall magnitude of shoreline diffusivity, with a secondary dependence on shoreline angle  $\theta$ , as demonstrated in calculations of the wave-climate averaged shoreline diffusivity for NCB (Fig. 4c)."

The caption for Fig. 4 was also updated accordingly:

"(c) Wave-climate-averaged shoreline diffusivity as a function of shoreline angle  $\theta$  calculated for a given  $\alpha$ , h,  $H_s$ , and T representative of NCB; vertical orange bar indicates the range of shoreline angles from the initial 2024 NCB shoreline."

No spectral domain calculation for waves was performed as long-term averaged wave statistics were taken from hindcast hourly wave conditions (L453):

"MEEB uses single representative values of  $\alpha$  and h for the entire shoreline, which, along with  $H_s$  and T, are derived from hindcast offshore wave conditions (described in Sect. 3.4)."

• Clarifying that alongshore variability in cross-shore sediment transport counteracts the tendency of alongshore diffusion to smooth the ocean shoreline into a straight line over time (L456):

"Therefore, the alongshore diffusion will tend to smooth the ocean shoreline towards a linear shape between the two endpoints of the domain over time, while alongshore variability in cross-shore sediment transport (e.g., overwash) counteracts this tendency by creating or sustaining perturbations in shoreline shape over time and can also move the endpoints in the cross-shore direction."

• Clarifying that model inputs in the probabilistic framework are exactly the same for duplicate simulations (L509):

"Duplicate simulations use the same exact model inputs, yet differ in their ecogeomorphic evolution because of the internal model stochasticity."

• Defining the total water level where first mentioned in the text and in the caption of Figure 4:

"...defined as an event in which the total water level (TWL; the sum of tide, surge, and wave runup) exceeds MHW." (L185)

"Total water level is the sum of tide, surge, and wave runup." (Fig. 4 caption)

• Including a plain-language description of the Brier Skill Score (L644):

"We define model performance primarily through the direct cell-by-cell comparison of simulated and observed elevation with the Brier Skill Score (BSS), which measures how much the simulated change improves a prediction relative to the baseline of predicting no change at all"

• Clarifying that a) during calibration of marine parameters, only the marine component of the model was utilized (all other components were inactive); and b) during calibration of the aeolian parameters, all components were utilized and active:

"First, Marine parameters alone were optimized using only the Marine component of MEEB over a single HWE event" (L661)

"Figure 5: Hindcast simulations testing performance of the MEEB Marine component with calibrated Marine parameters." (Fig. 5 caption)

"Results from the preceding marine parameter calibration were used to set the marine parameter values for the Aeolian calibration, and all components of MEEB (Aeolian, Marine, Shoreline, and Vegetation) were utilized and active." (L672)